

# Comparing global seismic tomography models using the varimax Principal Component Analysis

Olivier de Viron[1], Michel Van Camp[2], Alexia Grabkowiak[3], and Ana M. G. Ferreira[4,5]

[1]Littoral, Environnement et Sociétés (LIENSs UMR7266 La Rochelle University – CNRS)
[2]Royal Observatory of Belgium
[3]Institut de Physique du Globe de Paris, University Paris Diderot
[4]University College London
[5]Instituto Superior Técnico, Universidade de Lisboa

**Correspondence:** Olivier de Viron (olivier.de_viron@univ-lr.fr)

**Abstract.** Global seismic tomography has greatly progressed in the past decades, with many global Earth models being produced by different research groups. Objective, statistical methods are crucial for the quantitative interpretation of the large amount of information encapsulated by the models as well as for unbiased model comparisons. We propose here to use a rotated version of the Principal Component Analysis (PCA) to compress the information, in order to ease the geological inter-

pretation and model comparison. The method generates between 7 to 15 principal components (PC) for each of the seven tested global tomography models, capturing more than 97% of the total variance of the model. Each PC consists of a vertical profile, to which a horizontal pattern is associated by projection. The depth profiles and the horizontal patterns enable examining the key characteristics of the main components of the models. Most of the information in the models is associated with a few features: Large Low Shear Velocity Provinces (LLSVPs) in the lowermost mantle, subduction signals and low velocity anomalies likely

associated with mantle plumes in the upper and lower mantle, and ridges and cratons in the uppermost mantle. Importantly, all models highlight several independent components in the lower mantle that make between 36% and 69% of the total variance, depending on the model, which suggests that the lower mantle is more complex than traditionally assumed. Overall, we find that the varimax PCA is a useful additional tool for the quantitative comparison and interpretation of tomography models.

## 1 Introduction

Global seismic tomography has brought a new understanding of the current state of the mantle, by inversion of massive seismic data sets to build 3-D images of the Earth's interior, both of isotropic and anisotropic structure, the latter being one of the most direct ways to constrain mantle flow (e.g., Rawlinson et al., 2014; Chang et al., 2014; McNamara, 2019). The interpretation and comparison of tomography models often include computing correlations between two models with depth and degree, analyzing power spectra (e.g., Becker & Boschi (2002)), or visual inspections and qualitative or simple descriptions of the retrieved

patterns, for example of subducted slab or mantle plume candidates (e.g., Auer et al. (2014); French & Romanowicz (2014); Chang et al. (2016); Ferreira et al. (2019)). While the large-scale, upper mantle and lowermost mantle isotropic structure is fairly consistent from one model to the other, discrepancies appear when considering small-scale structures. Moreover, there



are substantial differences between existing global anisotropy models (e.g., Chang et al., 2014; Romanowicz & Wenk, 2017).
Nowadays, codes or web-based tools facilitate the interpretation and visual comparison of different models (e.g., Durand et al.
(2018), Hosseini et al. (2018)). This allows to identify regions with good agreement between seismic models using, e.g., vote
maps (Lekic et al., 2012) or through statistical tools showing the relative frequency of seismic anomalies at specific depth
ranges (Hosseini et al., 2018)). However, the large amount of information encapsulated in global tomography models, which
typically involve tens of thousands of model parameters, can be difficult to mine and to interpret efficiently.

Statistical methods used in other disciplines to analyse and classify big data and models may be useful to further enhance
the analysis of seismic tomography models, by providing a common ground for comparison. For example, in recent years,
clustering methods have been used to partition seismic tomography models into groups of similar velocity profiles, providing
an objective way of comparing the models (Lekic et al., 2012; Cottaar & Lekic, 2016). Here, we propose to use Principal
Component Analysis (PCA, Storch & Zwiers (1999)) to further explore this type of approaches. The PCA-based method aims
at approximating the tomographic models by a sum of a given number $\tilde{N}$ of components, with $\tilde{N}$ smaller than the actual
number of slices. Each PC consists of a vertical profile the principal component (PC) and a horizontal pattern, the load. Most
of the variance of the signal being captured by a reduced number of PCs, it allows to grab all the information by analysing only
the relevant components, resulting from an efficient compression.

The first PC, capturing the largest variance, often corresponds to an actual physical process, but the others are increasingly
difficult to interpret. The physical interpretation of the PCs and loads can be made easier by redistributing PCA components
along other eigen-vectors. In particular, the varimax criterion (Kaiser, 1958) allows focusing on PCs with large values concen-
trated on the smallest possible subset of depths, as it is physically likely that mantle structures have a limited depth extension
rather than spanning over the whole mantle depth. Previous studies in other fields of Earth sciences (see e.g. the thorough
review paper by Richman (1986)) showed that, when using the varimax criterion, the redistributed components are often less
sensitive to computation artifacts, for example related to data geometry.

The varimax analysis has previously been successfully used in various applications, such as to analyse climate models,
where the different models are projected on the same set of PCs, allowing a direct comparison in terms of capture variance
and retrieved features (Horel, 1981; Sengupta & Boyle, 1998; Storch & Zwiers, 1999; Tao et al., 2019; Kawamura, 1994).
Motivated by these successful results, in this study we apply the varimax PCA to the interpretation and comparison of global
tomography models. In section 2, we present the seven global tomography models used, followed by a description of the
statistical methods used in this study. Then, in section 4 we compare the classical and varimax PCA with a k-mean clustering
approach. Sections 5–6 present and discuss the results from the application of the varimax PCA to the seven tomography
models considered. We then propose a brief final discussion and conclusions in Section 7.

## 2 Seismic tomography models

We use seven 3-D global seismic tomography models: (i) S20RTS (Ritsema, 1999), 1999; (ii) S40RTS (Ritsema et al., 2011);
(iii) SEISGLOB2 (Durand et al., 2017); (iv) SEMUCB-WM1 (French & Romanowicz, 2014); (v) SGLOBE-rani (Chang et al.,





2015); (vi) S362WMANI+M (Moulik & Ekstrom, 2014); and, (vii) SAVANI (Auer et al., 2014). While the first three models are isotropic shear-wave speed models, the last four models also include lateral variations in radial anisotropy. These models were built from different data sets and using distinct modelling approaches, as summarised in Table 1. We focus on shear-wave models because the agreement of P-wave models is more limited (e.g., Cottaar & Lekic (2016)). The models used show the key features in current global isotropic and radially anisotropic models, and hence are representative of the current state of global tomography. For example, all isotropic shear-wave speed models show a good correlation with tectonic features in the upper mantle, such as mid-ocean ridges and cratons (see ∼ 100 km in Figure 1a). Moreover, they show the signature of subducting slabs around ∼ 600 km depth, as well as the two prominent large low shear velocity provinces (LLSVPs) beneath Africa and the Pacific in the bottom of the mantle at ∼ 2,900 km depth (Figure 1a). On the other hand, the agreement between the anisotropy models is much more limited (Figure 1b); common features between the models include a well-known positive radial anisotropy anomaly beneath the Pacific at ∼ 150 km depth, negative radial anisotropy anomalies beneath the East Pacific Rise at ∼ 200 km depth and negative radial anisotropy anomalies associated with the LLSVPs. The latter anomalies have been shown to be artefacts in the models due to the poor balance between SV- and SH-sensitive traveltime data in various existing body-wave data sets, which have much more data sensitive to SH- than to SV motions (e.g., Kustowski et al. (2008); Chang et al. (2014)). On the other hand, Moulik & Ekstrom (2014) showed that such spurious anisotropic features in even degree structure are reduced by using self-coupling normal mode splitting data in the inversions. Yet, trade-offs between isotropic and anisotropic structure in the lowermost mantle persist in odd degree structure, which is not constrained by self-coupling normal mode splitting data.




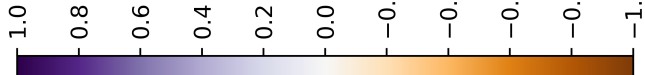









**Figure 1.** Depth slices of the isotropic and anisotropic models used in this study at the depths of 100, 200, 300, 400, 600, 800, 1000, 1400, 2000, and 2900 km. The isotropic (a) and anisotropic (b) models show perturbations in shear-wave speed and in $\xi = V_{SH}^2/V_{SV}^2$ with respect to PREM (Dziewonski & Anderson, 1981), respectively. The colour scale is normalised to vary from -max to +max, with the range of model amplitude variations shown at the left of each row. For simplicity, in the remainder of this paper we add "i" letters to the tomography models' names when referring to their isotropic part and we add "a" letters when referring to their radially anisotropic part; in the different figures "SGLOBE" will stand for SGLOBE-rani, and "SEMUCB" for SEMUCB-WM1

We obtained the global tomography models either directly from their authors or from the IRIS Earth model collaboration repository (REFS - http://ds.iris.edu/ds/products/emc/). Since some of the models have different reference 1-D models and use distinct parameterisations, for consistency we converted them into perturbations in shear-wave speed and in radial anisotropy with respect to the 1-D model PREM (Dziewonski & Anderson, 1981) on a common grid with a $1° \times 1°$ horizontal sampling and on 29 depth slices starting at 50 km depth and with a 100 km spacing from 100 km to 2900 km depth. In addition to the gridded representation allowing fair graphical presentation, we interpolate the data sets from the horizontal grids into an regular polyhedron of 9002 equiareal faces of which the vertexes were generated through the icosahedron tool of Zechmann (2019). This transformation produces an uniform sampling of the sphere, with each vertex having a surface corresponding to that of a $250 \times 250$ km² square. Considering that the statistical methods used in this study use the captured variance as a major criterion for ordering the components, the use of gridded data would overweight the contribution at the poles in the PC representation. The $180 \times 360 \times 29$ individual model data matrices are thus converted into $9002 \times 29$ matrices. As the shallowest layers contain the majority of the variability in the velocity anomalies, most of the principal components will be captured in those layers, which will be over-represented. Hence, the shear-wave speed and radial anisotropy perturbations are normalized by slice, i.e., the mean value of the slice is subtracted from each value, and each value is divided by the standard deviation of the slice. This is relevant as, in this study, we investigate relative values on a given profile; the actual magnitude can be recovered by multiplying the load patterns by the standard deviation of the layer in the original model. The normalisation applied to the models does not lead to a loss of information.

## 3 Methods

Previous studies have compared global tomography models using k-mean clustering (e.g., Lekic et al. (2012)). Though the PCA is very different on many aspects from the clustering of the k-mean method, it is useful to start by comparing PCA and varimax PCA results with those from the k-mean, for an illustrative tomography model (S40RTS).

### 3.1 k-mean clustering

Considering the three-dimensional data set $D(\lambda_i, \phi_i, z_j)$, the k-mean algorithm (MacQueen, 1967) defines $k$ clusters, corresponding to the $k$ sets of horizontal positions $(\lambda_i, \phi_i)$ closest to their average $z_i$ profiles. The algorithm is based on an iterative procedure. At the first iteration, it randomly chooses $k$ horizontal positions, used as cluster centers. Each point of the data set is then associated with the cluster center to which it is the closest to. The average radial profile of the points attributed to





| Model | Dataset | Parameterisation | Modelling approach |
|---|---|---|---|
| SGLOBE-rani (Chang et al., 2015) | Fundamental mode group-velocity data (T∼16-150 s), fundamental and overtone phase-velocity data (25-374 s, up to the 4th overtone), body-wave travel times. | Spherical harmonics up to degree 35 (laterally) and 21 spline functions (radially). 1-D reference model: PREM. | Ray theory. Regularisation: norm damping. |
| SAVANI (Auer et al., 2014) | Fundamental and overtone phase velocities (T∼25-370 s, up to the 4th overtone), body-wave travel times. | Variable size blocks on a $5° \times 5°$ base grid (laterally). 28 variable thickness depth layers. 1-D reference model: PREM. | Ray theory. Regularisation: vertical and horizontal smoothing. |
| S20RTS (Ritsema, 1999) | Fundamental mode and overtone phase velocities (T∼40-275 s, up to the 4th overtone), body-wave travel times, even-degree self-coupling normal mode splitting functions. | Spherical harmonics up to degree 20 (laterally) and 21 spline functions (radially). 1-D reference model: PREM. | Ray theory. Regularisation: norm damping. |
| S40RTS (Ritsema et al., 2011) | Expanded dataset of S20RTS (increased amount of measurements). | Spherical harmonics up to degree 40 (laterally) and 21 spline functions (radially). 1-D reference model: PREM. | Ray theory. Regularisation: norm damping. |
| SEMUCB-WM1 (French & Romanowicz, 2014) | Body waveforms (T > 32 s), surface waveforms (T > 60 s). | Spherical splines with spacing $< 2°$ (laterally) and 20 cubic b-splines (radially). 1-D reference model: own model. | Spectral element method for forward modeling, non-linear asymptotic coupling theory for inverse modelling. Regularisation: vertical and horizontal smoothing. |
| S362WMANI+M (Moulik & Ekstrom, 2014) | Fundamental mode phase velocities (T∼35-150 s), body wave traveltimes and waveforms, normal mode splitting functions. | 362 spherical splines (laterally) and 16 cubic splines (radially). 1-D reference model: own model. | Ray theory. Regularisation: vertical and horizontal smoothing. |
| SEISGLOB2 (Durand et al., 2017) | Fundamental and overtone phase velocities (T∼40-360 s, up to the 5th overtone), body-wave travel times, normal mode self- and cross-coupling coefficients. | Spherical harmonics up to degree 40 (laterally) and 21 spline functions (radially) | Ray theory. Regularisation: lateral smoothing controlled by an horizontal correlation length. |

**Table 1.** Global tomography models used in this study, including a short description of the data, parameterisation and the modelling approach used in their construction. All models were built using least-squares inversions with different regularisation choices.





each given cluster is computed and used as the new cluster center for the next iteration. This is repeated until convergence is achieved.

To make k-mean and PCA representations somewhat comparable, we use the clusters as horizontal patterns and the average vertical profiles of each cluster as the PCs. By construction, the variance captured by the k-mean is noticeably smaller than that

for the other methods, since it is not meant to propose a compressed representation of the data set, but rather to separate the data set into subsets, which results in an important loss of information.

### 3.2 Principal Component Analysis (PCA)

A 2D matrix $F_{j,k}, j = 1, 2, ..., J; k = 1, 2, ..., K$ is transformed by the PCA into a sum of components, each component being composed of a load $\alpha_{n,j}$ and an eigen-vector $A_{n,k}$.

$$F_{j,k} = \sum_{n=1}^{N=\min(J,K)} \alpha_{n,j} A_{n,k}.$$

In our case, $F_{j,k}$ corresponds to the velocity anomaly at horizontal position $p_j = (\lambda_j, \phi_j)$ and depth $z_k$. The $A_{n,k}$ are the eigen-vectors, or principal components (PC), of the covariance matrix. These PCs are orthogonal vertical structures representing the

covariance between the slices of the model. It has large positive values if the horizontal structures from two layers are positively correlated, zero values if the structures are not correlated, and large negative values if they are anti-correlated. The loading patterns $\alpha_{n,j}$ are also orthogonal to each other and each $\alpha_{i,j}$ results from the projection of the data set on the PC $i$, capturing the horizontal structure $i$ associated with the vertical anomaly profile $A_{i,k}$. The loads take continuous positive and negative values. Here, those patterns correspond to horizontal maps showing where each vertical structure is more or less important in

the model.

The components are ordered by decreasing eigen-value, as the variance captured by each PC is directly proportional to the eigen-value of the PC. Due to their orthogonality and to the mathematical properties of the transformation, the variance captured by each PC drops rapidly with the order, so that a small number of independent components ($\tilde{N} \ll N = 29$) is often sufficient to capture most of the information, allowing an efficient compression of the data set.

Unlike clustering methods, which are binary in that any horizontal location only belongs to one cluster, the PCA computes the amplitude of the contribution from each principal component, for every horizontal location, providing a compressed reconstruction of the data set.

The first PC corresponds to the dominant covariance, which might be physically associated with a global phenomena – in our case, a structure that would develop on the whole mantle depth – or a more local feature, i.e., associated with a limited depth

range. But this covariance structure might also correlate with other features from other depths, which will also be retrieved in the first PC. The second PC being orthogonal to the first, part of the physics might have been subtracted by the computation of the first PC, and it is even more so for the following principal components.





### 3.3 Varimax PCA

Following Neuhaus & Wrigley (1954), the physical interpretation of the PCs and load can be made easier by redistributing
PCA components along other eigen-vectors, by maximizing a functional of the loads that favors some physical properties that
appear physically meaningful (Storch & Zwiers, 1999). There are several possible redistribution options for the PCs (Storch &
Zwiers, 1999; Browne, 2001; Jolliffe, 2005). Among those, the varimax rotation (Kaiser, 1958) favors PCs with large values
concentrated on the smallest possible subset of depths and preserves the orthogonality of the PCs.

Considering the variance captured by the PCA components, the number $\tilde{N}$ of PCs to keep is selected to meet a given criterion:
the total variance captured, a fixed number of components, or the minimum variance captured by a PC kept. Then we define
new $\tilde{N}$ components, so that

$$F_{j,k} \simeq \sum_{n=1}^{\tilde{N}} \alpha_{n,j} A_{n,k} = \sum_{n=1}^{\tilde{N}} \beta_{n,j} B_{n,k}.$$

$\beta_{n,j}$ are the new horizontal structures and the new PCs, $B_{n,k}$, are chosen to maximize a given objective function $V$, defined by
the sum of the values of the objective functions $V_n$ computed over each PC:

$$V = \sum_{n}^{\tilde{N}} V_n \left( B_{n,k} \right),$$

with

$$V_n = \frac{1}{K} \sum_{k=1}^{K} \left( \frac{B_{n,k}}{s_k} \right)^4 - \frac{1}{K^2} \sum_{k=1}^{K} \left( \frac{B_{n,k}}{s_k} \right)^2.$$

where the $s_k$ are normalization factors, with $s_k = 1$, for all $k$ in the case of the varimax rotation.
This transformation corresponds to a rotation of the PCs because the subspace generated by the transformation – or the recon-
structed model – is the same as with the non-rotated PCA limited to $\tilde{N}$ components. In our case, it limits the vertical extension
of the PCs, i.e., each PC shows large values on only a few depths/slices.

The associated horizontal structures, $\beta_{n,j}$, are recomputed by projection of the tomography model on the rotated vertical
profile. This rotation conserves the orthogonality of the eigen-vector, which is not the case for all the possible rotations, but the
horizontal loads are often not orthogonal anymore (Mestas-Nuñez, 2000). The total variance captured by the rotated PCs is the
same as that from unrotated PCs, but the decrease of the variance is slower than that from the original decomposition.

## 4   Comparing the PCA, varimax rotated PCA, and k-mean results for the model S40RTS

Figure 2 shows the PCs and loads resulting from the application of the PCA, varimax rotated PCA, and k-mean clustering
methods to the tomography model S40RTS for a 6-PC decomposition. All the methods are applied on the same normalized
data.

Figure 2 highlights the complementary of the k-mean and varimax PCA methods. While the k-mean method allows to highlight
key large-scale features, such as e.g. the distribution of lowermost mantle velocity and to compare how it appears in the different





models (e.g., Lekic et al., 2012), the varimax-PCA approach provides a compressed representation of the full model. Being binary, the k-mean method does not provide amplitudes, i.e., the load for each PC is either 0 or 1: every location is part of one k-mean cluster, while it can be part of several PCs in the PCA analysis. The latter captures inherently more complexity with fewer principal components than k-mean clustering. Hence, it is not surprising that the six k-mean components capture only 34% of the total model variance, whereas both PC-based methods recover 83.2% (Figure 2). Therefore, while the k-mean method is a useful classification technique that allows a subset of data to be separated out from others, the varimax PCA is a distinct, valuable compression technique that reduces the number of parameters while minimizing loss of information. It allows identifying the most important components of tomographic models, easing their interpretation.

As suggested by Richman (1986), the PCA profiles $A_i(z)$ show increasingly oscillating patterns with $i$, which may lead to nonphysical interpretations. For example, the signature of ridges and cratons spread over the whole mantle (principal components 4 and 5, purple and cyan in the top row in Figure 2) observed in the unrotated PC representation makes no physical sense. More generally, the vertical profiles retrieved by PC analysis are certainly not all associated with sound geophysical structures. Only the first PC (red) provides directly interpretable patterns: the African and Pacific Large Low Shear Velocity Provinces (LLSVPs) (Garnero & McNamara, 2008; McNamara, 2019), whose depth extent with a maximum below 1800 km can be nicely visualised in Figure 2. On the other hand, even with this reduced number of 6 components, the varimax method recovers well-known structures such as ridges, cratons, oceanic plates, subduction zone and the LLSVPs (for a more detailed analysis see the next section). Based on these comparisons, we find that the varimax PC method is useful to concentrate at different depths coherent information that is available in the seismic tomography models, without any preconception. The next sections will thus focus on the application of this method to the interpretation of the global tomography models considered in this study.

## 5 Compressed information from varimax PCA

### 5.1 Comparison of vertical profiles and horizontal patterns

We use varimax PCA to compress the seven tomography models described in Section 2 into a set of components, keeping only the most important ones, as explained below. Each component is composed of a vertical profile obtained directly from the varimax process and of an associated horizontal pattern, which is computed by projecting the model on the profile. Such data compression is useful to compare the models if three major conditions hold. First, a subset of components must capture most of the variance of the signal, with the number of components being significantly smaller than the original number of depth splines/boxes, and enhancing the signal-to-noise ratio. Secondly, the relevant structures in the mantle, which will be used for comparison and for geological interpretation, should not be distorted by the compression, i.e., their shape and position must remain unchanged. Third, the power spectral densities should not be altered by the compression process.

In order to facilitate the comparison of the horizontal structures in the models, we label the varimax components obtained from the varimax PCA using capital letters in alphabetical order from components sensitive to shallow mantle structure to components sensitive to the lowermost mantle structure. Figure 3 shows the variance captured by the varimax PCA applied





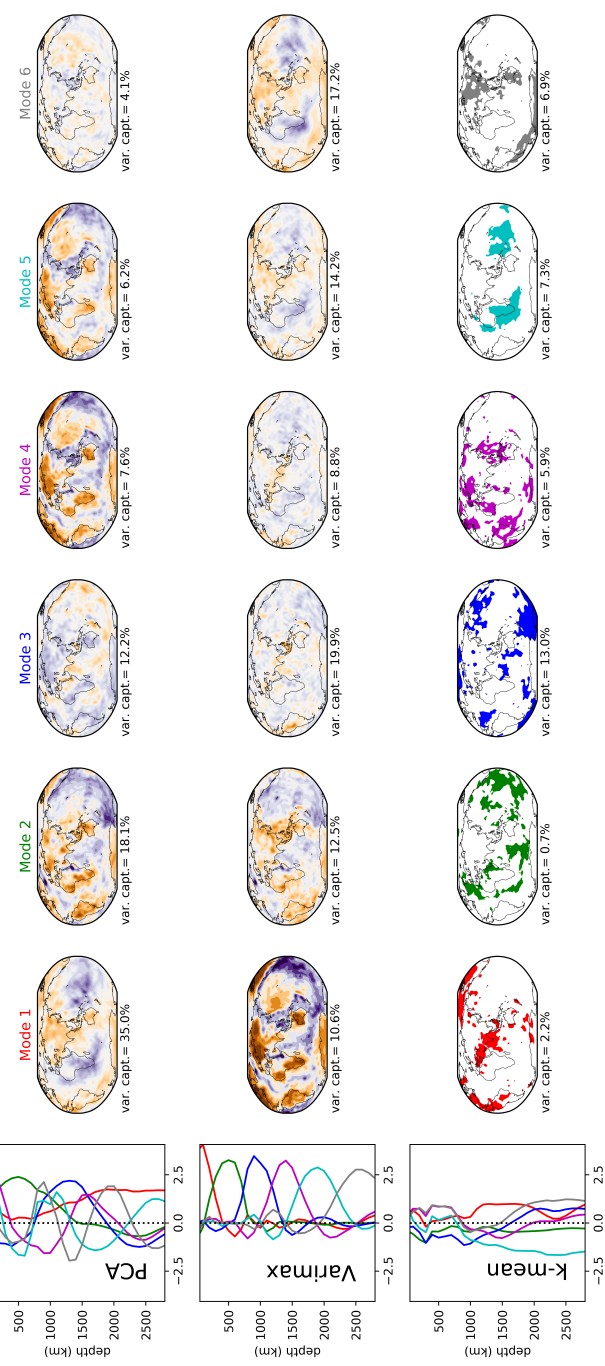

**Figure 2.** Results from the PC, varimax and k-mean methods applied to the model S40RTS, using 6 principal components. The varimax PC and loads are normalized in such a way that the horizontal patterns (load) range between -1 and 1, with orange corresponding to negative, blue to positive, and white to zero, and with the intensity being proportional to the value. We note that while this normalization eases the analysis, it does not lead to a loss of information (see main text for details). Unlike the classical variance-based sorting of the principal components, we order the varimax components by the depth of the profile's maximum.





to the different models. The varimax analysis is performed by considering all the components capturing more than 1% of the variance of the signal in the classical PCA. Keeping only the components with variance above 1% limits the number of maps, facilitating the quest for relevant information. Tests with 5% and 10% thresholds showed that some important information is lost. When using a 10% threshold, all the models are represented by 3 components only, which misrepresent known structures

such as ridges or subduction zones. Moreover, the depth distributions of the corresponding PCAs become quite broad and imprecise, each stretching over more than 1000 km depth ranges. In the 5% threshold case, the models are represented by 4 (SAVANIi) to 6 components (SEMUCB-WM1i, SEISGLOB2). Again, information is lost concerning e.g., ridges or subduction zones, and the depth information is spread over a depth range greater than 500 km for all components and for all models.

The simplification brought by the varimax method is particularly efficient for tomography models with weak regularization,

such as, e.g., SGLOBE-ranii, where short-scale structure is likely mixed with noise. Figure 3 shows that the number of varimax components ranges from 7 for model SAVANIi to 15 for SGLOBE-ranii, with the total variance captured by all these principal components always exceeding 97.3% (see also Table 2, which summarises the components kept in the varimax analysis). The number of varimax components required by each model depends on the details of the model's construction, such as e.g. the data used (Table 1) and, importantly, on subjective choices made, such as on the level of regularisation used. Increasing the strength

of regularisation reduces the model's effective number of free parameters and hence the number of varimax components required by the model. As the SAVANI model only needs 7 varimax PCA components, the shallowest component concentrates a lot of information (26% of the variance) that is spread into more components for the other models. Table 2 shows that the number of PCs needed to explain 97.3% or more of the total information in the tomography models is always smaller than the number of splines or layers used in the models' original depth parameterisation, with 29% to 75% fewer PCs than depth

splines/layers.

This fulfills the first condition for the usefulness of the data compression mentioned above. In order to check the second condition previously mentioned, Figure 4 compares 6 examples of depth slices in the original SEMUCB-WM1i model with those obtained from the model's reconstruction using varimax analysis. The differences between the original and reconstructed models are small and random, highlighting that the compression process does not distort the model's features. Finally, we also

verified that there are only very small, random differences between the power spectra of the original and of the reconstructed models (see Figures A1 and A2 in the supplementary material). Hence, the third condition of usefulness of the data compression used in this study is also satisfied.

Figure 5 shows the varimax PCs for all the tomography models used in this study, together with the spline functions or the variable thickness depth layers used in the models' parameterisation. The vertical profiles differ from one model to the other

both in numbers of components and in the depths of their maxima. For example, as shown in Figure 3, SAVANIi requires 7 components, while SGLOBE-ranii needs 15 components. We re-emphasise that the number of principal components obtained for a given model reflects their amount of independent information, which in turn depends heavily on choices made during the model's construction, such as regularisation.

The horizontal patterns obtained from the varimax PCA also show distinct features, but this is not always in the same way as

for the vertical profiles (Figures 6, B1-B6). S20RTS shows sharp vertical profiles (Figure B3), and contains even one more



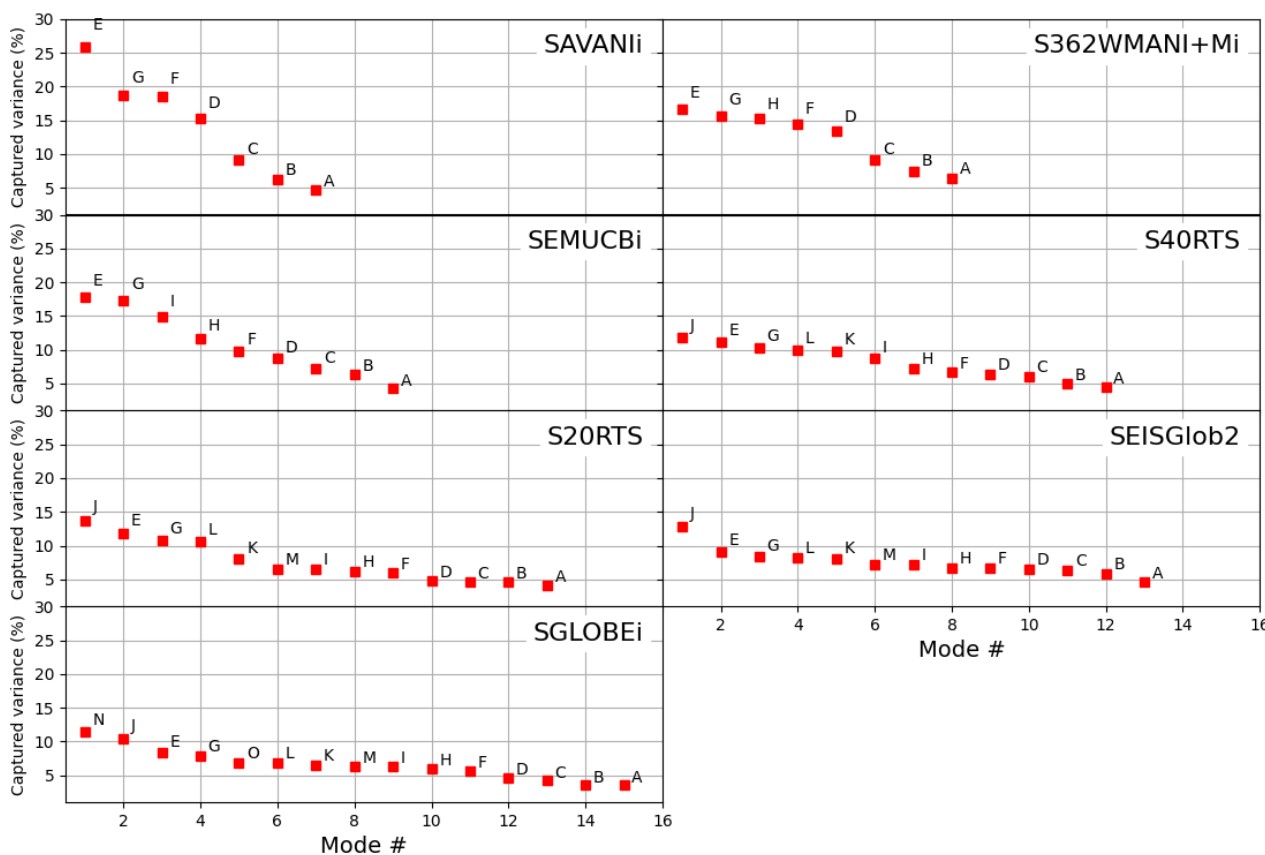

**Figure 3.** Captured variance by the PC varimax method, applied to the isotropic part of the seven tomography models used in this study. The number of components is chosen such that during the PCA, we only keep the components explaining more than 1% of the variance, which occurs after 7 (SAVANIi) to 15 (SGLOBE-ranii) components. The varimax PCs are sorted alphabetically from the shallowest one to the deepest ones.

PC (13) in the upper mantle than the updated S40RTS model (12, Figure B4), but the horizontal patterns are smoother. This is likely due to the fact that the latter model is constrained by about 10 times more data and used a different level of regularisation (Ritsema et al., 2011). SGLOBE-ranii, SEMUCB-WM1i and SEISGLOB2 (Figures B1, 6 and B6) depict sharper horizontal patterns than SAVANIi, S362WMANI+Mi and S20RTS (Figures B2, B5 and B3). SEMUCB-WM1i and SAVANIi (Figures 6 and B2) show vertical profiles concentrated closer to the surface, but their horizontal patterns are different. The PC B of SAVANIi (∼200 km depth, Figure B2) corresponds to low velocity anomalies underneath all oceans, which is also the case for S20RTS and S362WMANI+Mi (Figures B3 and B5), but not for SGLOBE-ranii and SEISGLOB2 (Figures B1 and B6). Such upper mantle low velocities beneath the oceans also appear in the models S40RTS and SEMUCB-WM1i, but with a higher level of detail (Figures B4 and 6).




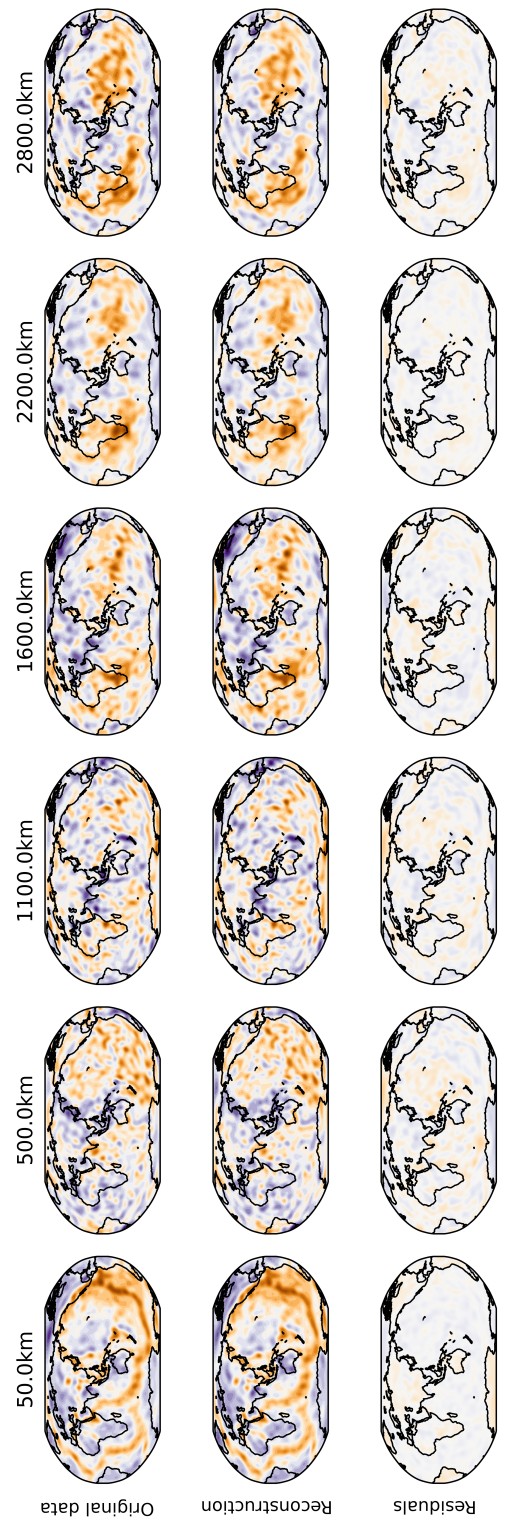

**Figure 4.** Examples of 6 depth slices in the original SEMUCB-WM1i model (top), in the model recovered without the principal components with less than 1% of the variance, i.e., with 9 components (middle), and differences between the two (bottom).



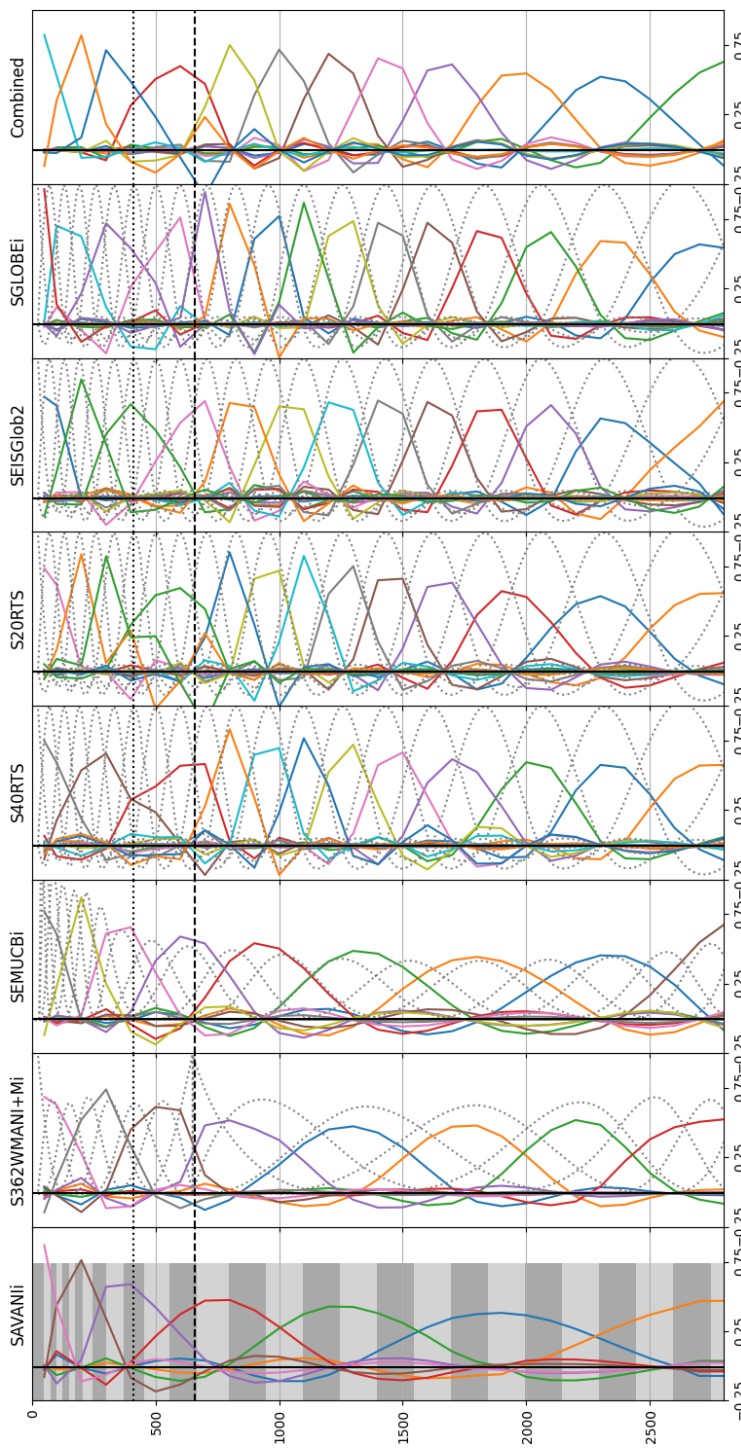

**Figure 5.** Principal components for the different individual model varimax-PCA and the combined one (see section 6), for the isotropic part of the 7 tomography models used in this study. Only the PC components above 1% are kept in this analysis (Figure 3). The dashed grey lines represent the spline functions and the grey boxes the variable thickness depth layers used by the different models. The vertical dashed lines indicate the 410 and 660 km seismic discontinuities.





We note on Figure 5 that the vertical components obtained from the varimax analysis do not fully correspond to the depth parameterisations used to build the models, especially above the 660 km discontinuity, where there are 2.3 to 3.3 lesser PCs than original spline functions or boxes. This implies that the PCs do not just simply reflect the model parameterisation and inform us about the independence between the slices reconstructed from the model. In the upper mantle, for all models, we end up with 3 to 4 PCs. In the lower mantle, the correspondence differs from one model to the other, whereas we observe three

categories:

1. SGLOBE-rani, where the PCA reproduces quite well the original spline functions except the two deepest ones, which are recovered into one PCA. This is probably due to the relatively weak regularisation used (Chang et al., 2015).

2. SEISGLOB2, S20RTS and S40RTS, for which most of the PCs reproduce the splines. For S20RTS and S40RTS, one PC encompasses the depth associated with two splines between 1500 and 2000 km, while the first spline just underneath the

660 km discontinuity is not taken over by any PC. For SEISGLOB2 there is one PC for two splines between 800 and 1000 km.

3. S362WMANI+Mi, SAVANI and SEMUCB-WM1i: 10 splines are encompassed by 5 modes for SEMUCB-WM1i and 8 splines by 5 modes for S362WMANI+Mi. For SAVANIi, the structure of the PCs seems independent from the box parameterisation.

This shows that overall the tomography models do not have a strong imprint of the depth regularisation used in their construction. This is especially true above the 600 km discontinuity or in the whole mantle for the SAVANIi, S362WMANI+Mi and SEMUCB-WM1i modelS, where the splines or boxes seem over-sample the available information.

One of the most striking differences between the models is the way the signal is distributed between 500 and 1,500 km depth. In this region the different tomography models require between 2 (SAVANIi [D-E]) and 7 PCs (3 for S362WMANI+Mi [C-E] and

SEMUCB-WM1i [D-F]; 5 for SEISGLOB2 [D-H]; 6 for S20RTS [D-I] and S40RTS [C-H]; and 7 for SGLOBE-ranii [D-J]). The observed variability in the number of required PCs likely reflects the level of model regularisation used in the construction of the various tomography models. The PC E of SAVANIi (∼1200 km depth) is dominated by low-velocity anomalies and shows a substantially different pattern to e.g. the PC F of SEMUCB-WM1i and the PC G of SEISGLOB2 (∼1300 km depth), which depict alternating low and high velocity zones. The principal components G (∼1000 km depth) and H (∼1100 km depth)

of SGLOBE-ranii present mostly low velocity anomalies, which are similar to the components F of S40RTS and G of S20RTS, both at ∼1100 km depth, but in these two latter models we also observe a high-velocity anomaly under the north-west Indian ocean. In SAVANIi, this is also observed on its PC E (∼1200 km depth), which is however much more broadly distributed at depth. These differences between the models reflect the high level of uncertainty for this part of the mantle, which is likely due to its limited data coverage.

## 5.2 Geophysical interpretation

A fully detailed geological and geophysical discussion of the models is beyond the scope of this study, and has already been performed in many previous studies (see e.g., McNamara (2019); Flament et al. (2017); Ballmer et al. (2015); Pavlis et al.





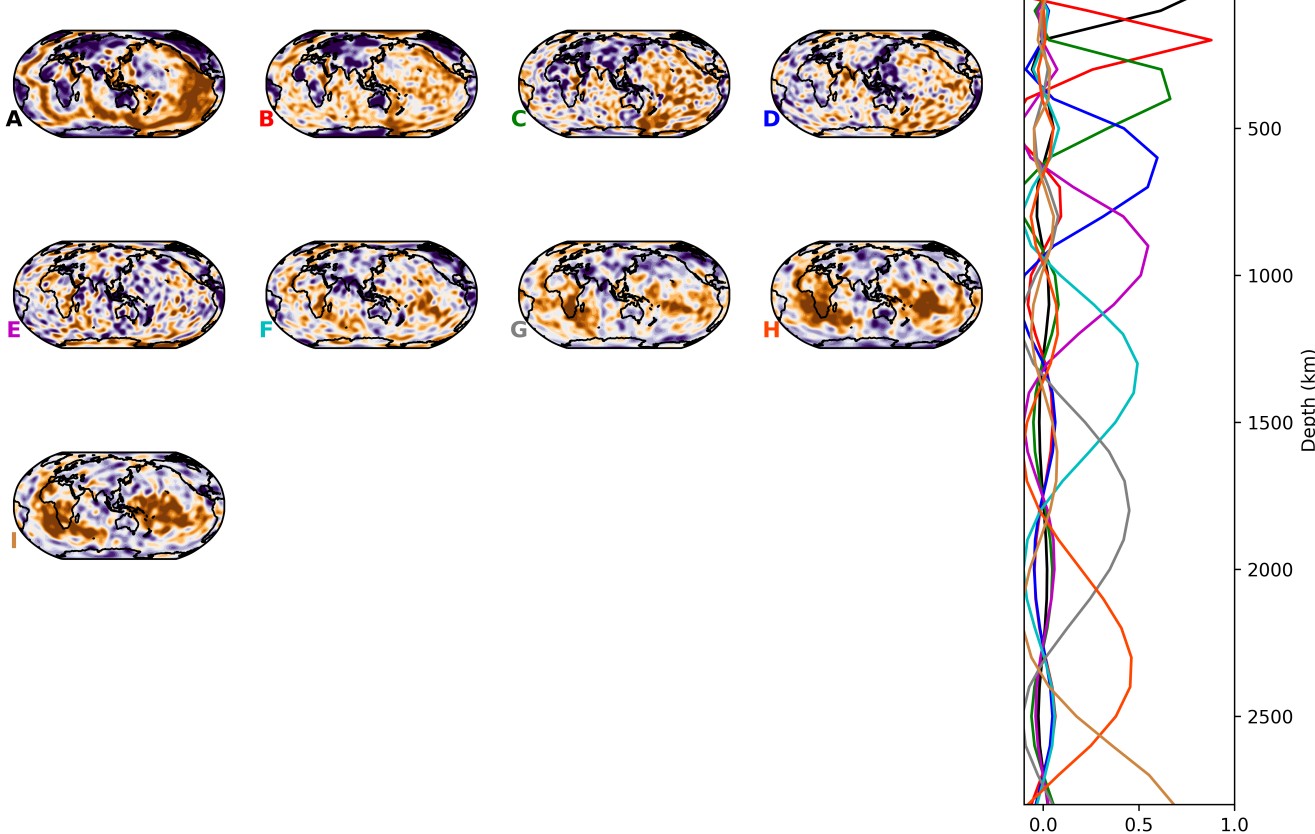

**Figure 6.** 9 Varimax components of the SEMUCB-WM1i model. On the right, the principal components or vertical profiles, and on the left, the associated horizontal structures. The other models are shown in the supplementary material on Figures B1-B5.





(2012); van der Meer et al. (2018); Rudolph et al. (2015); Ritsema et al. (2011)). The varimax PCA recovers all the features discussed in these studies. Table 3 compares how key Earth structures are captured by the varimax analysis for the isotropic
part of the seven tomography models used in this study (Figures 6, B1 - B5). The table covers ridges, rifts, plateaus (low-velocity anomalies, red) and cratons (high-velocity anomalies, blue) at depths of 50 – 300 km, subducted slabs (high-velocity anomalies) at 300 – 1300 km depth and the LLSVPs and the Perm low velocity zone in the lowermost mantle (red anomalies). Depending on the regions, the high velocity craton signature should reach a maximum depth between 100 and 175 km (Begg et al., 2009; Heintz et al., 2005; Polet & Anderson, 1995), but tomography models often show a deeper signature, likely due to
smearing effects. The model SGLOBE-ranii is the most consistent with this depth limit, as most of the cratons are concentrated in the second PC (∼100 km depth, PC B on Figure B1), separating them from the cold oceanic crust. This is possibly due to the huge set of data sensitive to the upper mantle used in SGLOBE-ranii's construction, including massive sets of both phase and group velocity measurements. Beneath Africa and the Baltic region, a high velocity zone remains visible on the third PC (∼300 km depth, PC C on Figure B1). For the other models, the craton signatures extend from 50 to ∼200-300 km depth.

The low velocity zones underneath the Tibetan plateau (Legendre et al., 2015) and Hangai dome, south-west of the Baikal lake (Chen et al., 2015), are recovered by the first PC of all models with a maximum at 50 km, but with different shapes. These zones are smaller in SEMUCB-WM1i (Figure 6) than in SGLOBE-ranii and SAVANIi (Figures B1 and B2), and the Tibetan plateau extends more to the south in SAVANIi and S362WMANI+Mi (Figures B2 and B5), being subdivided into 3 small zones in S40RTS and SEMUCB-WM1i (Figures B4 and 6).

From ∼150 km to ∼800 km depth, global tomography models often show a low velocity anomaly beneath the Pacific (e.g., Lebedev & van der Hilst, 2008). For SAVANIi, at ∼200 km depth, it is difficult to distinguish between the Pacific and other oceans. Its PC C, with a maximum at ∼400 km depth, is confined to the central and western Pacific, while the PC C of S362WMANI+Mi, at ∼500 km depth, is more to the south west, resembling the PC D of S20RTS (∼600 km depth), D of S40RTS (∼700 km depth), E of SGLOBE-ranii (∼600 km depth), D of SEMUCB-WM1i (∼600 km depth) and D of
SEISGLOB2 (∼700 km depth).

All models show a high velocity zone between ∼300 and 700 km depth beneath the central Atlantic and along the Atlantic coasts of South America and Africa, similar to that from Figure 9 of Ritsema et al. (2011), and already discussed in Ritsema et al. (2004). Nevertheless, this zone appears less clearly in SEMUCB-WM1i (Figure 6 [C-D]) and especially SGLOBE-ranii (Figure B1 [C-D]), where it is mixed with low velocity patches. This anomaly is a region with long transform faults, high
gravity, anomalous ocean depth, and low melt production. It is thought to be the region of the Atlantic that formed during the final stages of the opening of the Atlantic because it was presumably the strongest part of the Pangean continent (Bonatti, 1996).

In the East African rift, the low velocity anomaly aligned with the Afar Depression and the Main Ethiopian Rift in the uppermost mantle (Benoit et al., 2006), Hansen & Nyblade (2013) appears in all the models, from the surface to the LLSVP, with narrower
contours in S40RTS, SEMUCB-WM1i, and SGLOBE-ranii than for the other models. This is consistent with the presence of one or of multiple mantle plumes in the region, as proposed in previous studies (e.g., Hansen et al. (2012); Chang & Van der Lee (2011); Chang et al. (2020)).





All models show high-velocity subduction zones in the western Pacific, among others, notably underneath the Philippine plate over two principal components with depths ∼400-800 km. This complex system mixes different subduction zones (van der

Meer et al., 2018). The Izu-Bonin slab subducts westerward down to ∼870 km depth and is connected in the upper ∼300-400 km depth to the Marianas to the south, which plunges vertically down to ∼1200 km depth (components [C-E] in Figure 6, [C-G] in Figure B1, [C-D] in Figure B2, [D-F] in Figure B3, [C-D] in Figure B4, [C-D] in Figure B5, and [C-F] in Figure B6) More to the south, north of Papua New Guinea, the Caroline Ridge, from ∼475 to 750 km, and, west of those zones, Manila, Sangihe, and even more west, Banda, Sumatra, and Burma, also present high velocity anomalies. This is recovered

by SGLOBE-ranii (components [D-E] in Figure B1), SEMUCB-WM1i (components D in Fig. Figure 6) and SEISGLOB2 (components D in Figure B6), but is broader, especially to the north, in S20RTS (components [D-E] in Figure B3), S40RTS (components C in Figure B4), SAVANIi (components [C-D] in Figure B2), and S362WMANI+Mi (components C in Figure B5).

The Tonga-Kermadec subduction zone, located below the south Fiji Basin down to a depth of ∼1300 km in the lower mantle

(van der Meer et al., 2018), is recovered by all models, but is less clear in S362WMANI+Mi (Figure B5). Conversely, SGLOBE-ranii, S40RTS, SEISGLOB2 and to a lower extent, SEMUCB-WM1i, show a narrow arc-shaped signature of this zone. On the other hand, it is difficult to assess a maximum depth of this subduction zone in SGLOBE-ranii.

All models evidence the LLSVPs, though they are less clear in some models, such as SAVANIi and S362WMANI+Mi (Figure B5), and they appear quite patchy in S40RTS and SGLOBE-ranii (Figures B4 and B1). All the models show low velocity

anomalies spreading from the core-mantle boundary (CMB), where the LLSVPs are clearly visible, to about 1500 km depth, where low-velocity structures are less coherent (for example, components G-I in Figure B1). All together, the components encompassing the LLSVPs capture 11% (SAVANIi) to 29% (S40RTS) of the models' information. The Perm anomaly is recovered in all models for the two deepest components, apart for SAVANIi, where it is recovered by the last PC only. This is because this PC is quite broad, extending from ∼2000 km depth to the CMB. These depths are consistent with, e.g., the findings

of Lekic et al. (2012) and Flament et al. (2017), which estimate that LLSVPs spread up to ∼500 km above the CMB. Note that it is difficult to estimate the top of the LLSVPs on S362WMANI+Mi, as there are persistent low velociy zones beneath e.g. eastern Europe up to the PC E centered at ∼1300 km depth (Figure B5)

Our analysis allows determining the importance of the various elements of the models. For example, for all models, principal components with maxima in their varimax PCs below 1,700 km depth and dominated by LLSVPs explain 11% (SAVANIi) to

24% (SEISGLOB2) of the model's information. On the other hand, principal components with maxima in the top 300 km and dominated by ridges, rifts and cratons explain 22% (SGLOBE-ranii) to 45% (SAVANIi) of the model's information.

## 6 Combined PCA

The horizontal patterns associated with each PC result from the projection of the tomography model on the varimax PCs, which differ from one model to the other. As suggested by Sengupta & Boyle (1998) in another context, it is interesting to

compare the different models using a common PCA, which removes the inconsistencies between the representations. Thus, we



| Model | # splines or boxes | Single (#components) | Combined (12 components) |
|---|---|---|---|
| SGLOBE-ranii | 21 | 98.2 (15) | 92.0 |
| SAVANIi | 28 | 98.4 (7) | 98.9 |
| S20RTS | 21 | 98.1 (13) | 95.4 |
| S40RTS | 21 | 97.3 (12) | 96.2 |
| SEMUCB-WM1i | 20 | 97.9 (9) | 97.7 |
| S362WMANI+Mi | 16 | 98.5 (8) | 97.9 |
| SEISGLOB2 | 21 | 97.7 (13) | 94.7 |

**Table 2.** Variance [%] obtained from the individual varimax analysis of each model. In parenthesis, the number of components capturing more than 1% of the variance is shown. The last column provides the variance captured by 12 components, for the combined analysis of the 7 models, as discussed in Section 6. The second column provides the number of splines or boxes originaly used in the models.

apply a varimax PC analysis to the seven models stacked on the horizontal axis, i.e., to a $7 \times 9002 = 63014$ by 29 matrix, and refer to the results as a combined analysis in the remainder of this paper. Using the same 1% threshold limit as used before, this analysis generates 12 varimax PCs, i.e., 12 vertical profiles (A-L) common to the seven models. Then, we compute the horizontal structures associated with each PC by projecting each of the 7 models on those vertical profiles (Figures 7 and D1
to D11).

Components A-D (at ~50, 200, 300, 600 km depths) are mostly confined in the upper mantle, while lower mantle structure is represented by components E-L (~800, 1000, 1200, 1400, 1700, 2000, 2300, 2800 km depths), as shown in the vertical profiles from the combined varimax analysis presented on the last column of Figure 5. As expected, the vertical profiles from the combined analysis are smoother than those from the individual model analysis for the more detailed models (SGLOBE-
ranii, SEMUCB-WM1i, S40RTS and SEISGLOB2), and sharper for the smoothest tomography models (SAVANIi, S20RTS, SEMUCB-WM1i and S362WMANI+Mi). Note that the 1% criterion is applied globally, and not on the individual models, as was done in the previous section. The last column of Table 2 shows the total variance captured by the 12 components for each model. It shows that the combined analysis is very efficient for the smooth SAVANIi model, capturing 98.9% of its variance, whereas the variance captured for the other models lies between 92.0% and 97.9%. SGLOBE-ranii model is only resolved at
the 92.0% level, which is not surprising as it is more detailed than the other models (likely due to the use of less regularisation), with the individual analysis requiring 15 PCs and allowing a finer localisation of the models' patterns (Figure B1).

Most of the patterns described in Table 3 and discussed in the previous section are also recovered by the combined analysis. This common projection makes it easier to compare the components E (~800 km depth) to G (~1200 km depth) in the lower mantle. These components (Figures 7 and D5–D6) capture ~20.1% of the information in the models and display a similar pattern in
SGLOBE-ranii, S20RTS, S40RTS and SAVANIi. On the other hand, SEMUCB-WM1i, S362WMANI+Mi and SEISGLOB2 show different features, such as e.g. fewer high-velocity zones in this depth range. Regarding principal component H (~1400 km depth, Figure D7), it describes 7.2% of the information. All the models show a similar large-scale pattern except for S362WMANI+Mi, which shows isolated low-velocity zones in the Pacific, especially in the south.



| Model | SGLOBE-ranii | SAVANIi | S20RTS | S40RTS | SEMUCB-WM1i | S362W MANI+Mi | SEIS GLOB2 |
|---|---|---|---|---|---|---|---|
| # of components | 15 | 7 | 13 | 12 | 9 | 8 | 13 |
| **Ridges, rifts and plateaus** | | | | | | | |
| African Rift LV[1] | ✓ | ✓ | ✓ | ✓ | ✓ | ✓ | ✓ |
| Fast spreading Pacific zone | 50-100 | 50-200 | 50 | 50 | 50 | 50 | 50 |
| Tibetan plateau, Hangai dome LV[2] | 50 | 50 | 50 | 50 | 50 | 50 | 50 |
| **Craton HV zones** | | | | | | | |
| African | 100-200 | 50-200 | 50-300 | 50-300 | 50-200 | 50-300 | 50-200 |
| Antarctic | 100 | 50-200 | 50-300 | 50-300 | 50-200 | 50-300 | 50-200 |
| Arabian | 100 | 50-200 | 50-300 | 50-300 | 50-200 | 50-300 | 200 |
| Australian | 100 | 50-200 | 50-300 | 50-300 | 50-200 | 50-300 | 50-400 |
| Baltic | 100-200 | 50-200 | 50-300 | 50-300 | 50-200 | 50-300 | 50-200 |
| Siberia | 100 | 50-200 | 50-300 | 50-300 | 50-200 | 50-300 | 50-200 |
| Indian | 100 | 50-200 | 50-300 | 50-300 | 50-200 | 50-300 | 50 |
| North American | 100 | 50-200 | 50-200 | 50-300 | 50-200 | 50-300 | 50-400 |
| South American | 100 | 50-200 | 50-200 | 50-300 | 50-200 | 50-300 | 50-200 |
| **Back arc LV** | | | | | | | |
| Japan | 50-100 | 50-200 | 50 | 50 | 50-200 | 50 | 50 |
| Philippines | 50-100 | 50-200 | 50 | 50 | 50-200 | 50 | 50 |
| Tonga-Kermadec | 50-100 | 50-200 | 50 | 50 | 50-200 | 50 | 50 |
| **Subducted slabs** | | | | | | | |
| Izu Bonin-Mariana HV | 300 | 200 | 200 | 300 Poor | 400 | ? | 400 |
| East Pacific HV | 600-700 | 400-800 | 600-800 | 700 | 600 | 500 | 700 |
| Tonga-Kermadec HV[3] | 100-1300 | 400-1200 | 600-1300 | 800-1500 | 400-1300 | 300-1800? | 200-1000 |
| North Pacific, Sunda HV[3] | 800-1100 | 400-1200 | 1000-1300 | 1000-1100 | ? | 500-800? | 400-1000 |
| **Others** | | | | | | | |
| Pacific LV | 300-600 | 200-400 | 200-300 | 300 | 200-600 | 300-500 | 200-700 |
| Central Atlantic HV[4] | - | 400 | 300-600 | 300-700 | 400-600 | 500 | 400-700 |
| **Lower mantle structures** | | | | | | | |
| LLSVPs | 1400-2700 | 1900-2800 | 1500-2800 | 1500-2800 | 1300-2800 | 1800-2800 | 1900-2800 |
| Perm LV | 2300-2700 | 2800 | 2300-2800 | 2000-2800 | 2300-2800 | 2200-2800 | 2100 |

**Table 3.** Examples of key geophysical patterns recovered in the mantle by the varimax analysis (see Figures 6 and B1 to B6). HV = high velocity zone, LV = low velocity.

[1]No clear interruption from the surface down to the CMB; [2]Legendre et al. (2015); Chen et al. (2015); [3]van der Meer et al. (2018); [4]Ritsema et al. (2011).





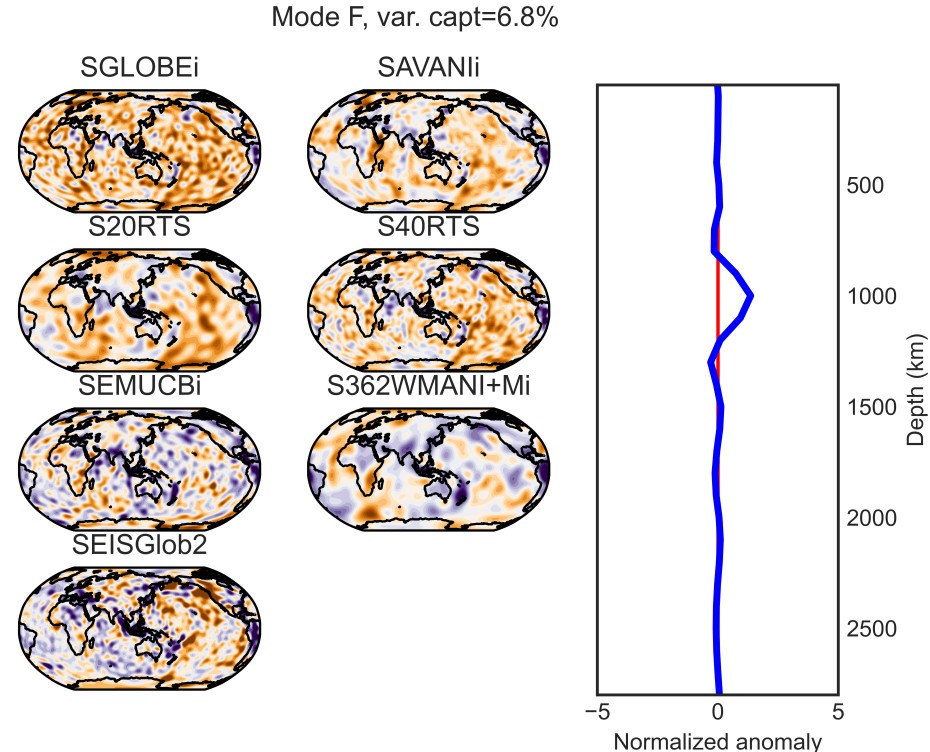

**Figure 7.** PC F (maximum at 1000 km depth) of the combined analysis of the isotropic models. On the right, the principal components or vertical profiles, and on the left, the associated horizontal structures.The other components are shown in the supplementary material on Figures D1 to D11.

As shown in the previous section, all models display several independent components in the lower mantle, making between

36% (SAVANIi) and 69% (SEISGLOB2) of the total components, depending on the model. This highlights complexity in the lower mantle and supports recent studies suggesting that the region of the lower mantle above the lowermost D" layer is more complex than previously thought. For example, slab stagnation and lateral deflection of mantle plumes have been proposed in the uppermost lower mantle (Fukao & Obayashi, 2013; French & Romanowicz, 2015). Moreover, intriguing observations of seismic discontinuities (Kawakatsu & Niu, 1994; Jenkins et al., 2017) and of scatterers (Kaneshima, 2016) have also been

reported at these depths. Compositional layering (Ballmer et al., 2015), a viscosity increase (Marquardt & Miyagi, 2015; Rudolph et al., 2015), and spin transitions that seem to occur in Fe-bearing mantle minerals (Lin et al., 2013) have been proposed in the lower mantle, which can potentially influence the region's elasticity and transport properties.





## 6.1 Anisotropic structure

In addition to isotropic shear-wave speed anomalies, four of the models considered in our study also include radial anisotropy
perturbations, that is, speed differences between vertically and horizontally polarised shear waves: SGLOBE-rania, SEMUCB-
WM1a, SAVANIa, and S362WMANI+Ma. The seismic imaging of anisotropy is more challenging than that of isotropic struc-
ture because the sensitivity of seismic data to anisotropy is weaker (e.g., Chang et al., 2014; Beghein & Trampert, 2004;
Romanowicz & Wenk, 2017). Moreover, it has been shown that if crustal effects are not properly modelled, this can lead to
substantial errors in the estimated mantle anisotropy (Panning et al., 2010; Ferreira et al., 2010; Chang et al., 2016; Bozda? &
Trampert, 2008; Leki? et al., 2010). These difficulties are at least partly responsible for the strong differences between existing
radial anisotropy models (e.g., Figure 1b). Varimax PC analysis is thus a natural candidate to analyse and compare the models
since it enhances their robust information. Figures 8 and E1–E9 show the vertical profiles (varimax PCs) and horizontal patterns
from the combined varimax analysis on the anisotropic part of the four radially anisotropic models, for which 10 components
capture more than 1% of variance. As expected, there is poorer agreement between the radial anisotropy structure in the models
than between the isotropic structure discussed in the previous sections, though some common features can be identified.

The two LLSVPs appear on the deepest PC J of SGLOBE-rania and SEMUCB-WM1a (Figure E9), which captures 10% of
the models' information. The Pacific LLSVP appears barely in the last PC for SAVANIa and S362WMANI+Ma. However, as
explained in section 3, e.g. Kustowski et al. (2008) and Chang et al. (2014, 2015) showed that the signature of LLSVPs in
radial anisotropy models is an artefact due to leakage of isotropic structure into artificial anisotropic structure in the lowermost
mantle.

For PC B (with a maximum at ∼100 km depth, Figure E2), a positive zone appears beneath the Pacific and the Nazca
plates in SGLOBE-rania and SAVANIa, and to a lesser extend in SEMUCB-WM1a, while no clear pattern is evidenced in
S362WMANI+Ma. The same holds true for mid-ocean ridges. A positive anomaly is observed on PC C under the Pacific plate
for all models with a maximum around a depth of 200 km (Figure 8). A broad positive radial anisotropy anomaly beneath the
Pacific at these upper mantle depths has been well documented in previous studies, and may be due to horizontal mantle flow
in the region and/or thin layers of partial melt in the asthenosphere (e.g., Ekstr'om & Dziewonski (1998); Gung et al. (2003)).
Components B (maximum depth 100 km, Figure E2), D (maximum depth 400 km, Figure E3) and especially C (Figure 8)
evidence subduction patterns on SGLOBE-rania (Alaska, Izu-Bonin, Fiji-Tonga-Kermadec). This is also observed, but less
clearly, for SAVANIa and S362WMANI+Ma on PC C. We also distinguish subduction signatures deeper in the mantle on PC
F along Cascadia, Central and South America, Tonga, the western Pacific and the north of the Mediterranean Sea (Figure E5).
An overall red negative anisotropy anomaly for the first PC A is common to SEMUCB-WM1a and SAVANIa (Figure E1).
SGLOBE-rani shows such red anomaly only under the oceans, which is probably due to the different way the crust is treated in
this model, with crustal thickness perturbations being jointly inverted for along with isotropic and anisotropic structure (Chang
et al., 2015).





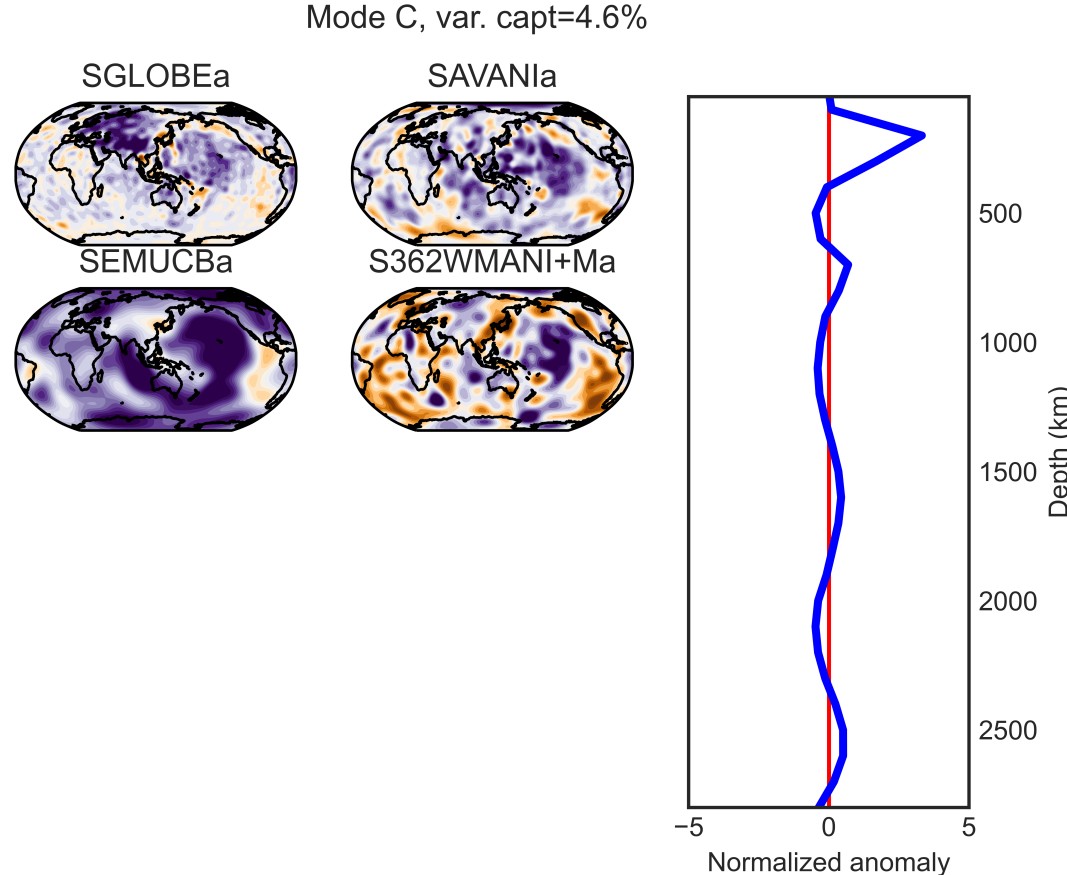

**Figure 8.** PC C (maximum at 200 km depth) of the combined analysis of the anisotropic models. On the right, the principal components or vertical profiles, and on the left, the associated horizontal structures. The other components are shown in the supplementary material on Figures E1 to E9.

## 7 Conclusions

Discussion and Conclusion

Global seismic tomography models typically involve thousands to tens of thousands of parameters, which can be cumbersome to handle and difficult to interpret. This is also true for model comparison, where we lack a common basis for comparing models built with different parameterisations. In this study we used a rotated version of the Principal Component Analysis to compress the information, to ease the geological interpretation and model comparison. The varimax PC analysis results in a separation of the information into different components associated with depth distributions, which are linked to a horizontal pattern obtained by orthogonal projection. We tested the analysis on seven global tomography models: S20RTS, S40RTS,





SEISGLOB2, SEMUCB-WM1, SAVANI, SGLOBE-rani, and S362WMANI+M, where the latter four include laterally varying radial anisotropy. We analysed the models both individually as well as jointly.

We found that using the varimax method we reduce by 29% to 75% the number of independent depth components needed to describe more than 97% of the total information in the tomography models. Moreover, the model compression process did not lead to any significant loss of information. Hence, the varimax analysis simplified the number of patterns that needs to be analyzed and by ensuring the orthogonality of the depth components it eased the detection and comparison of the relevant information. Overall, the large majority of depth components and horizontal maps obtained from the varimax analysis are

different from the original parameterisations used for building the models. This is especially true above the 600 km discontinuity or in the whole mantle for the SAVANIi, S362WMANI+Mi and SEMUCB-WM1i modelS, where the spline functions over-sample the available information. This implies that the PCs do not only reflect the model parameterisation and inform us about the independence between the slices reconstructed from the model. This also shows that the tomography models do not have a strong imprint of their model parameterisation. The varimax technique, being a data compression method, allows an

easier view of the whole information present in the tomography models, and is complementary to clustering methods, such as the k-mean technique, which allow to evidence zones of comparable properties. Both can help users get a better understanding of the complex Earth's interior structure.

When applying the varimax analysis to isotropic tomography models, we found that the most important elements of the models contributing to most of the information are: (i) Large Low Shear Velocity Provinces (LLSVPs) in the lowermost mantle; (ii)

subducted slabs and low velocity anomalies probably associated with mantle plumes in the upper and lower mantle; and, (iii) ridges and cratons in the uppermost upper mantle. The analysis highlights several independent components in the lower mantle that make between 36% and 69% of the total components depending on the model, which supports recent studies suggesting that the lower mantle is more complex than previously thought. The reasons of this complexity remain a very active field of research. On the other hand, we find limited agreement between the radial anisotropy structure of the models, with common

features mainly in the asthenosphere and to some extent in the lower mantle beneath the Pacific and beneath subduction zones. Choices such as data types and the strength of regularisation used in the construction of tomography models are probably key controls of the number of varimax components required by each model. Hence, the PCA-based model compression preserves the impact of the choices made in the construction of the tomography models, and facilitates their interpretation in terms of geophysical objects. Future work will focus on comparisons with other sources of information, such as e.g. gravity anomalies,

magnetic anomalies or heat fluxes.

*Code and data availability.* A python function for computing PCA and varimax PCA is provided at https://gitlab.univ-lr.fr/odeviron/ varimax/. With our configuration, i.e., 29 slices with 9000 points per profile, it analyses an individual model in about 0.15 CPU sec, whereas the combined analysis with 7 models takes about 1.3 CPU sec, using a 2.3 GHz 8 cores Intel Core i9.





## Appendix A: PSDs

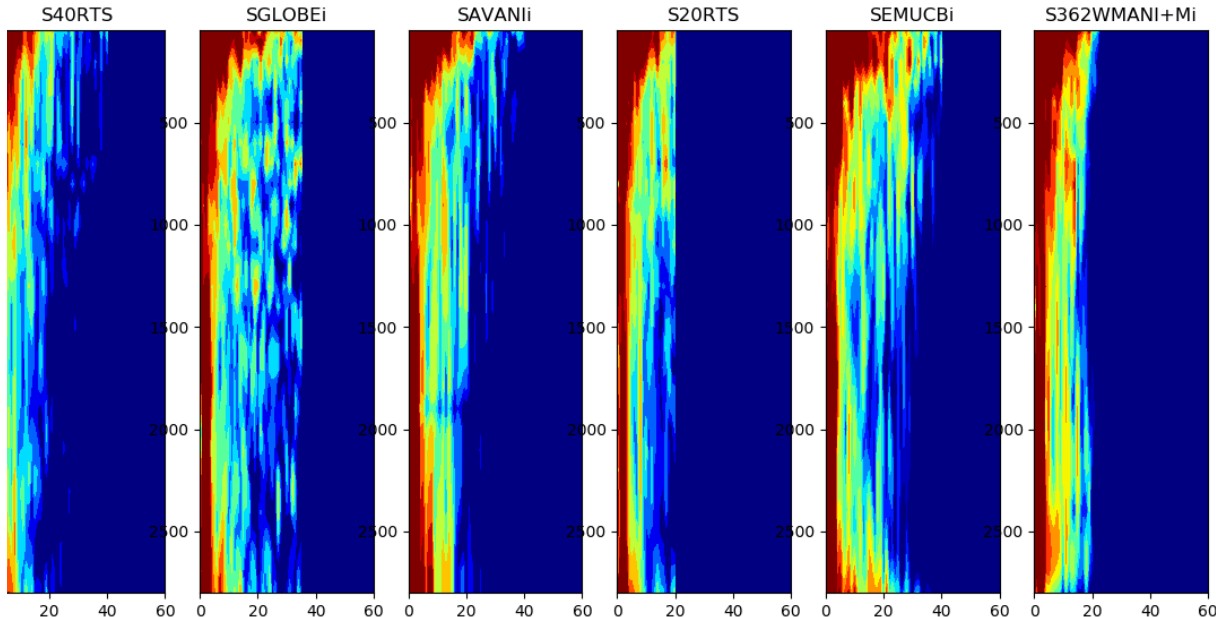

**Figure A1.** PSDs of the original models.

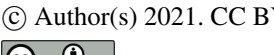


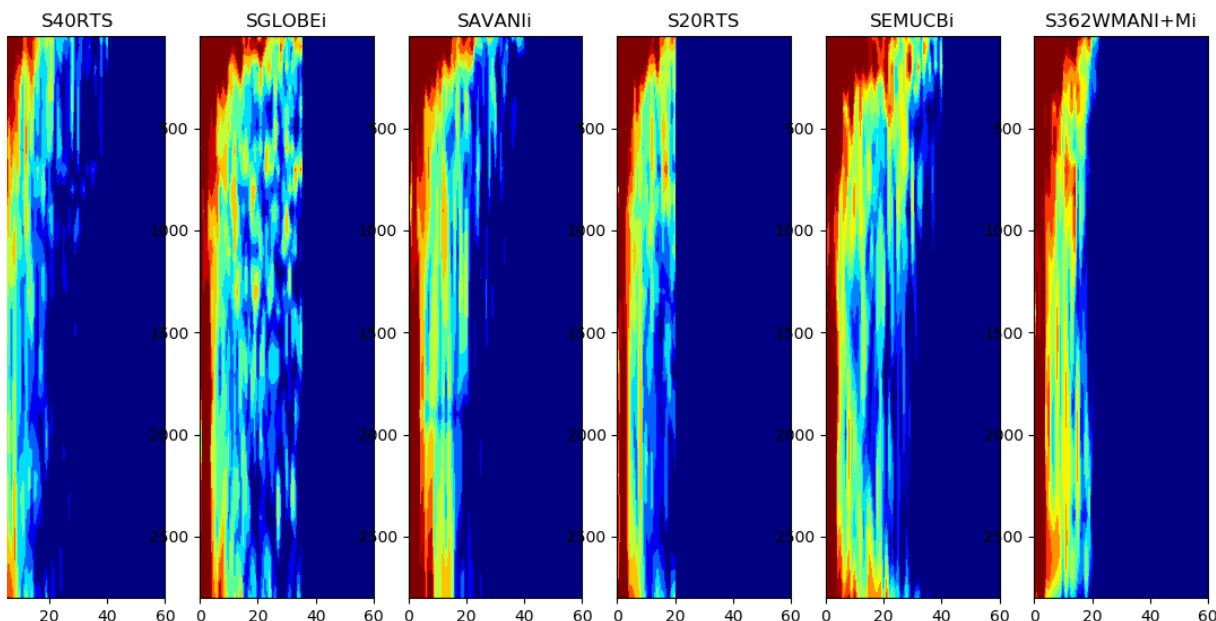

**Figure A2.** PSDs of the reconstructed models, from the varimax modes. The number of modes is given in Table 2.



**Appendix B:  Varimax PCA**

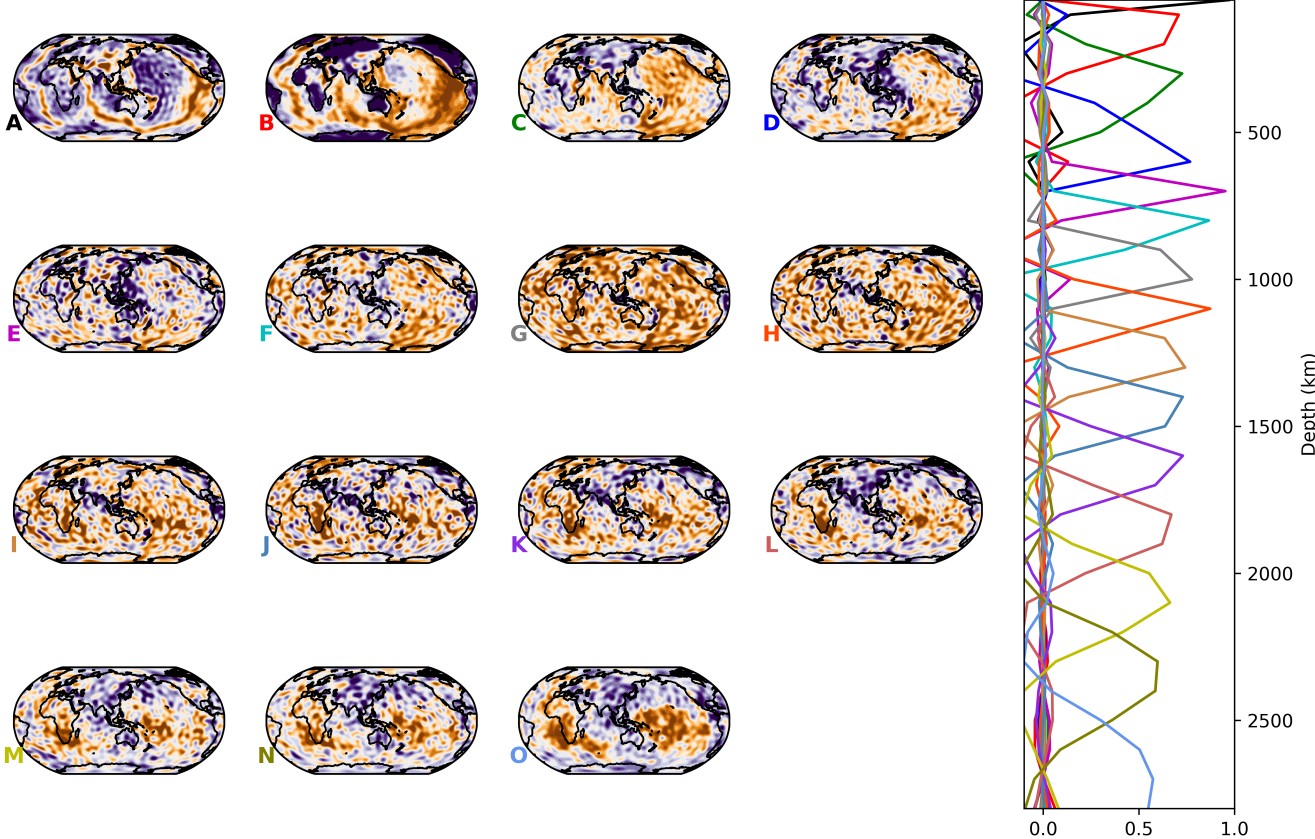

**Figure B1.** 15 Varimax components of the isotropic part of the SGLOBE-rani model.



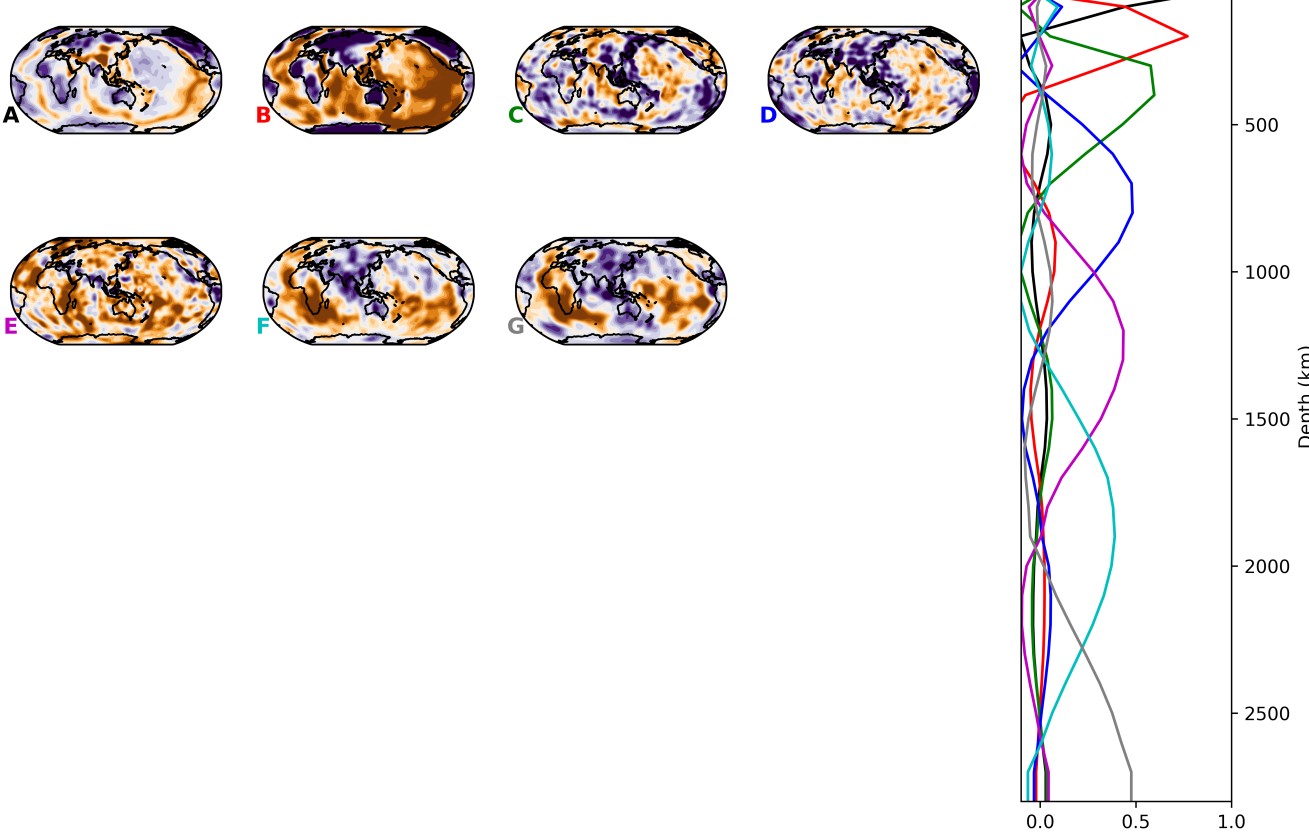

**Figure B2.** 7 Varimax components of the isotropic part of the SAVANI model.



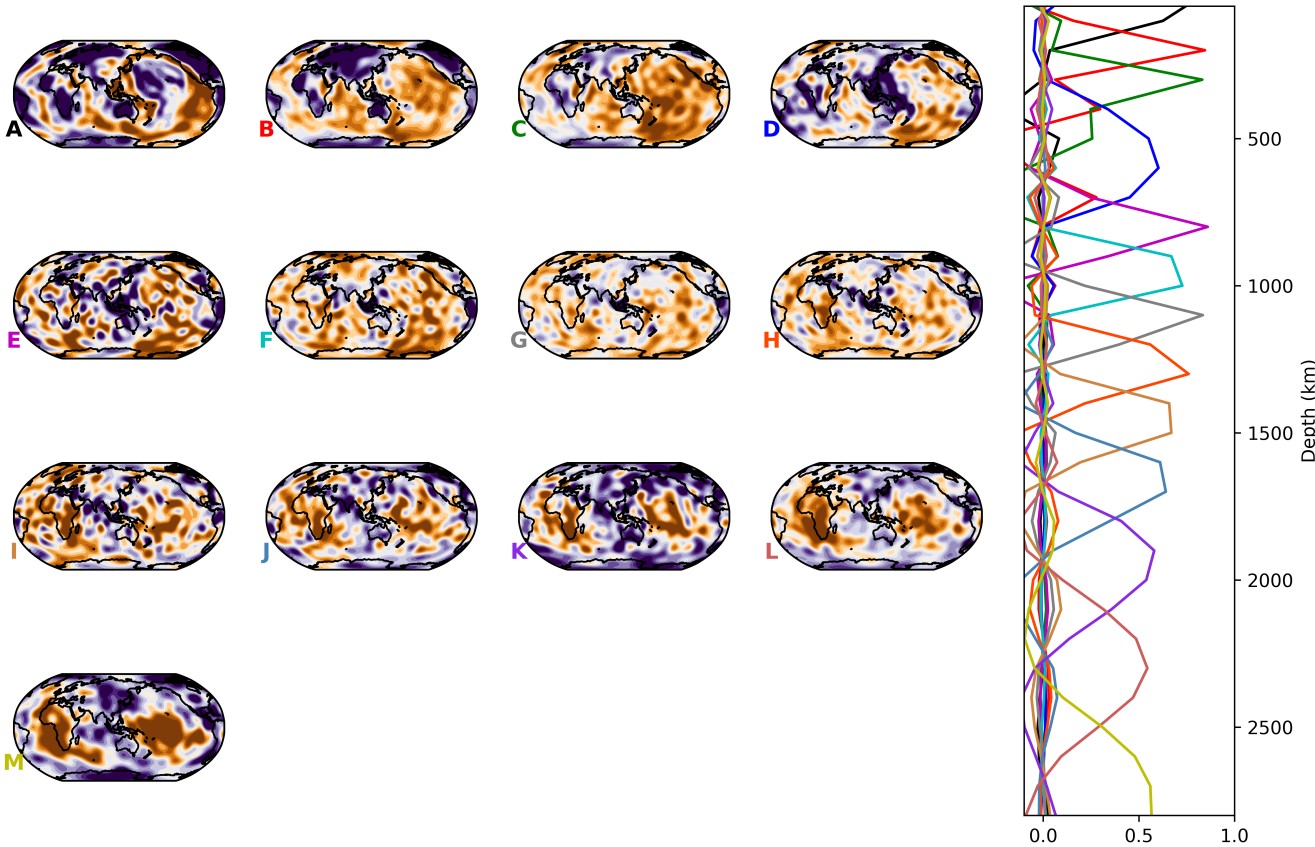

**Figure B3.** 13 varimax components of the S20RTS model.





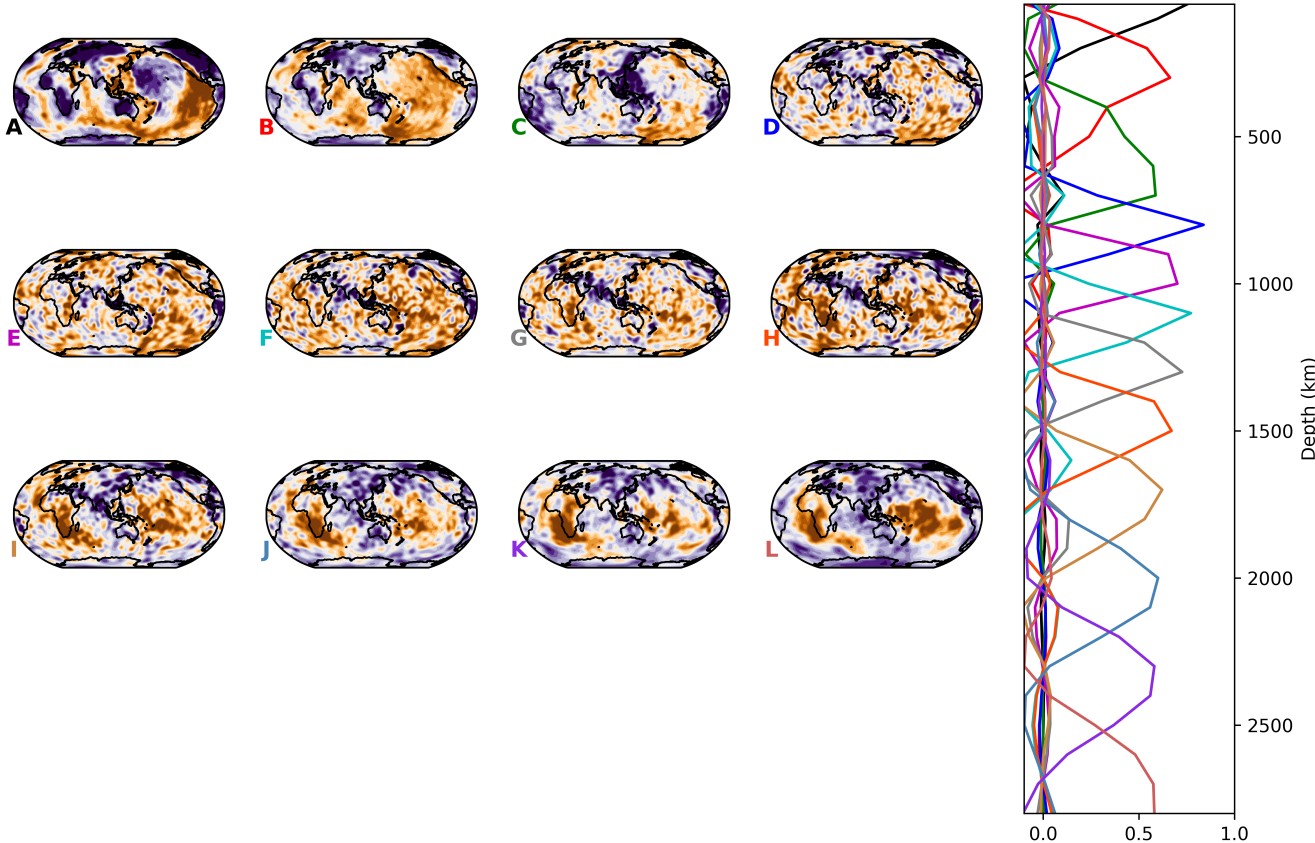

**Figure B4.** 12 Varimax components of the S40RTS model.



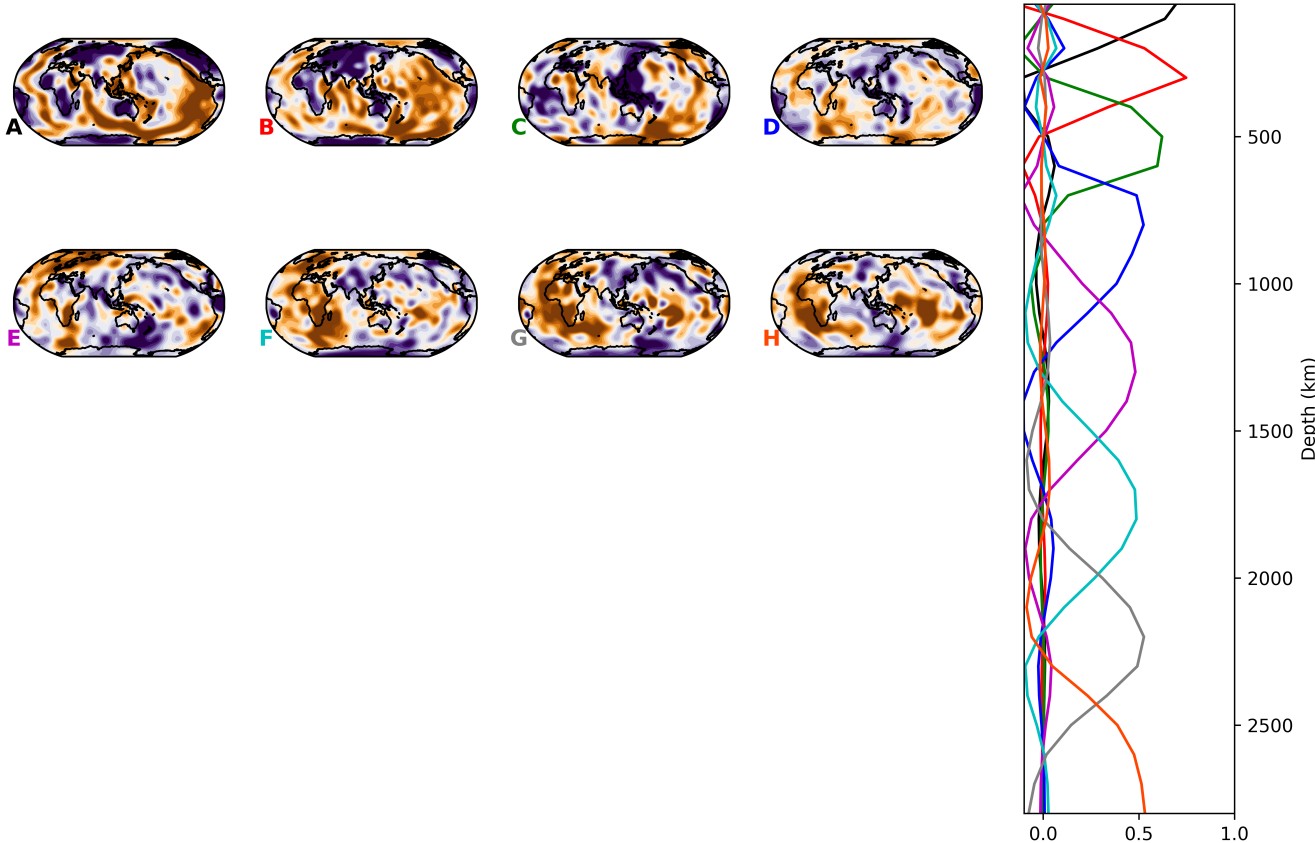

**Figure B5.** 8 Varimax components of the isotropic part of the S362WMANI+M model.



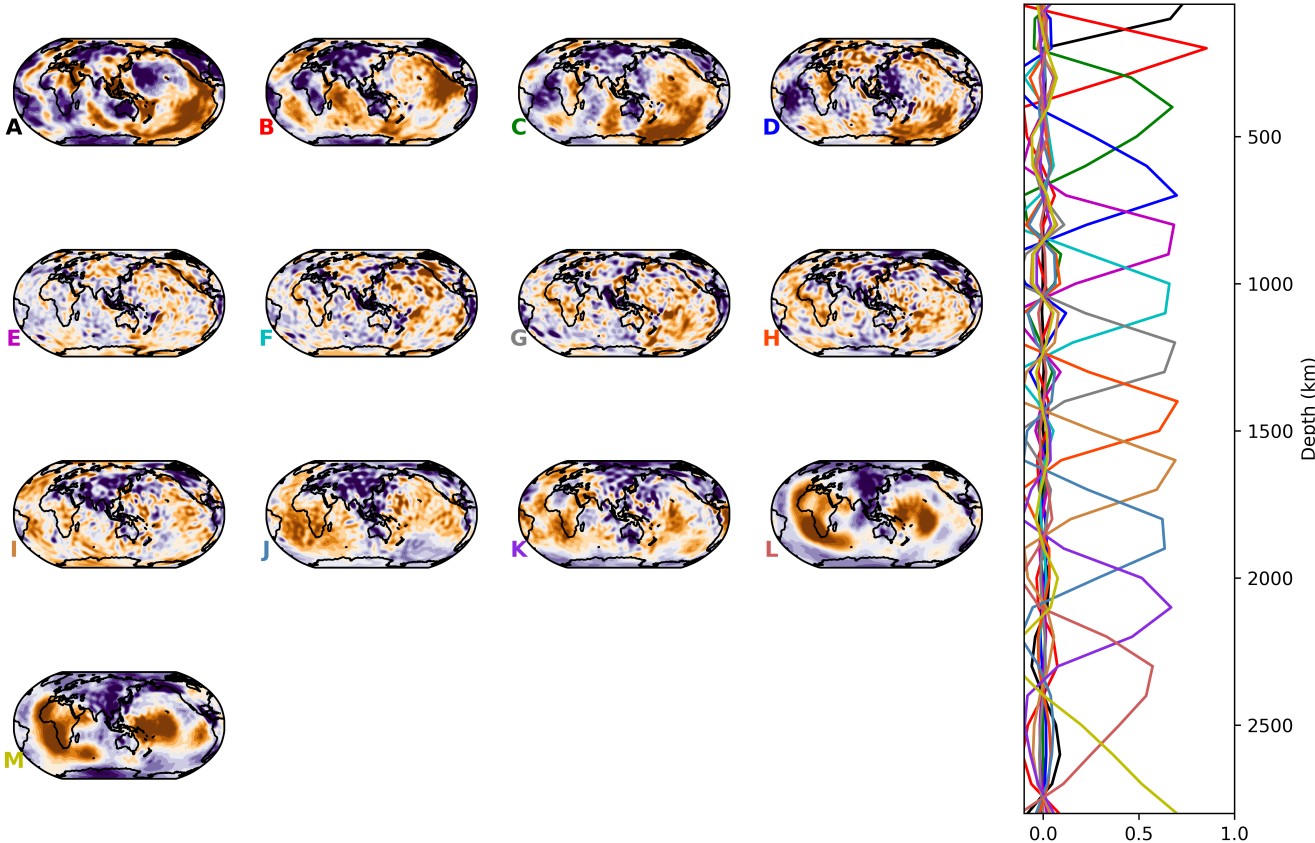

**Figure B6.** 13 Varimax components of the isotropic part of the SEISGlob2_R model.




**Appendix C: Combined PCA**

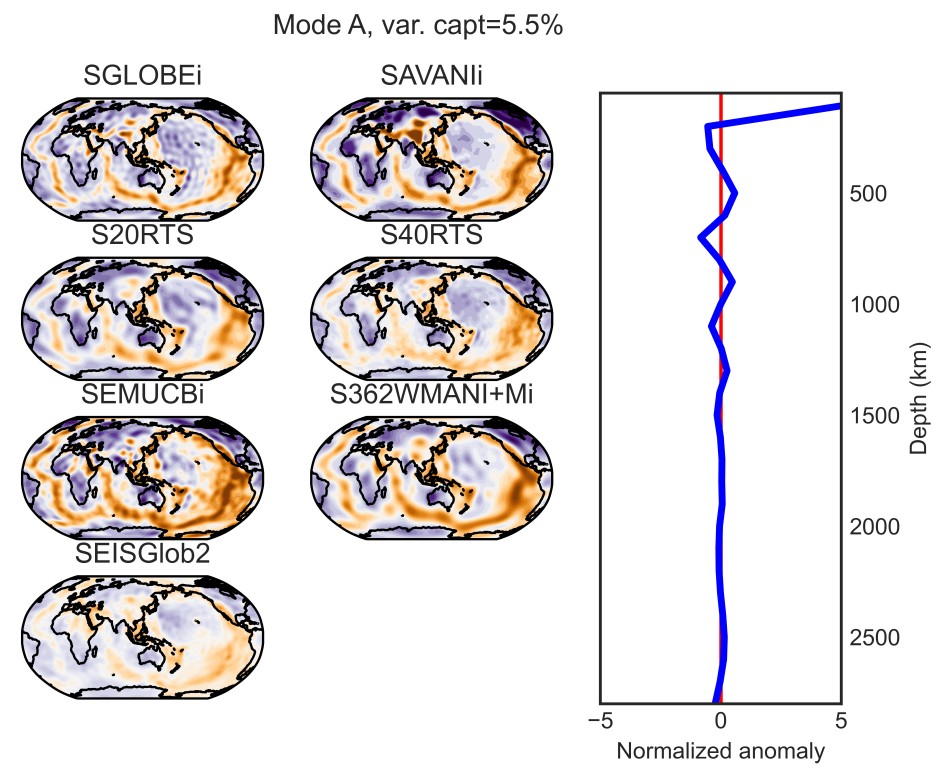

**Figure C1.** PC A (maximum at 50 km depth) of the combined analysis of the isotropic parts of the models.



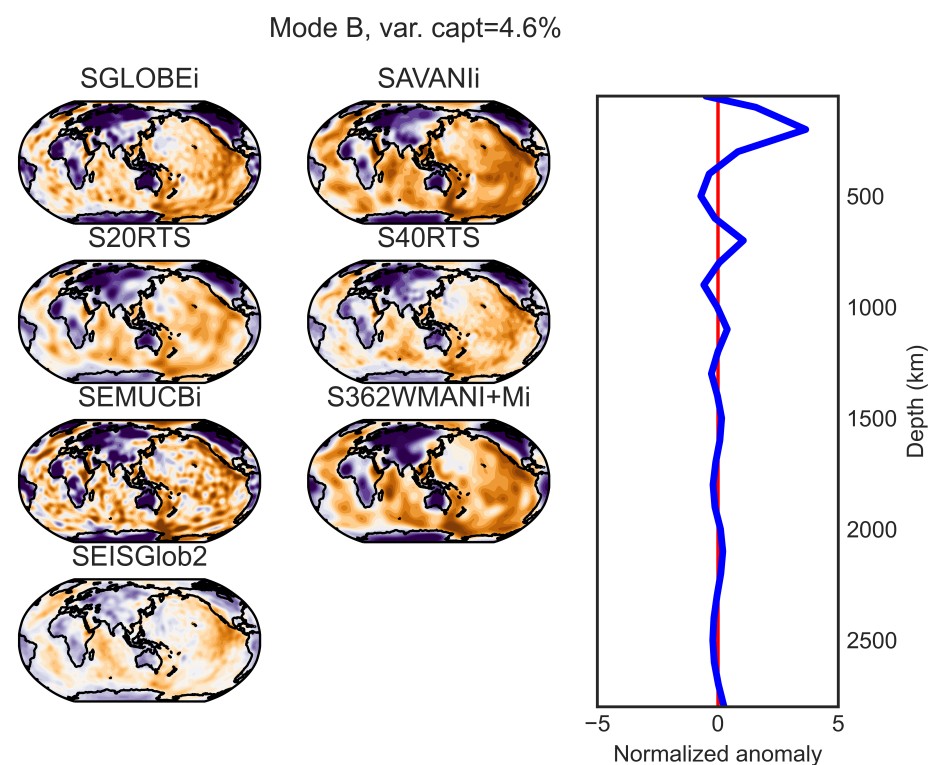

**Figure C2.** PC B (maximum at 200 km depth) of the combined analysis of the isotropic parts of the models.





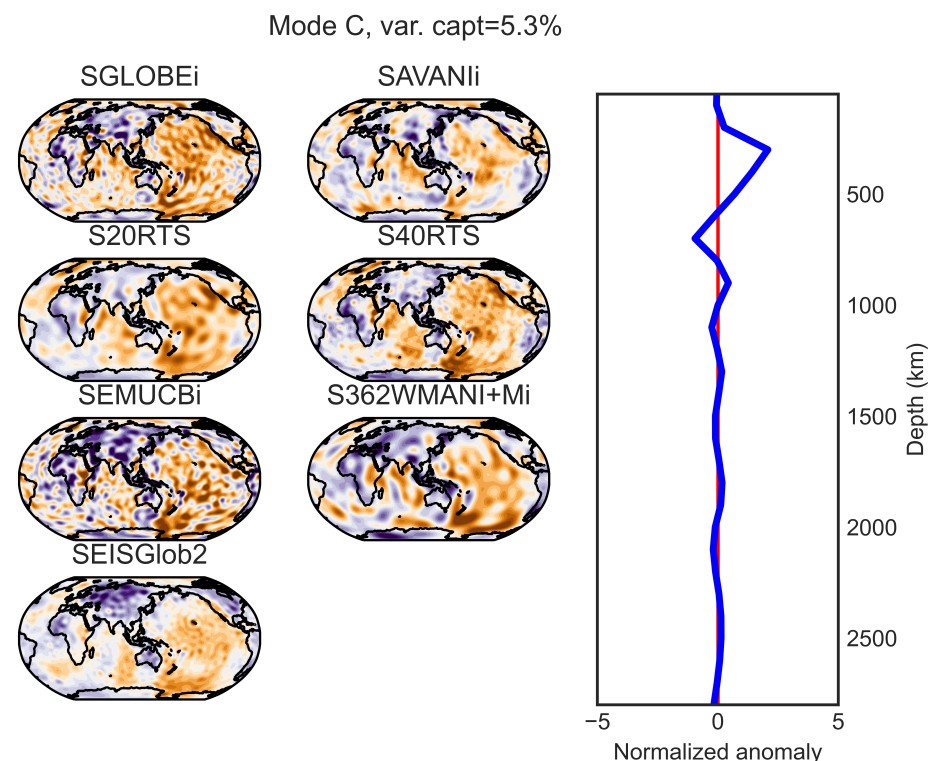

**Figure C3.** PC C (maximum at 300 km depth) of the combined analysis of the isotropic parts of the models.

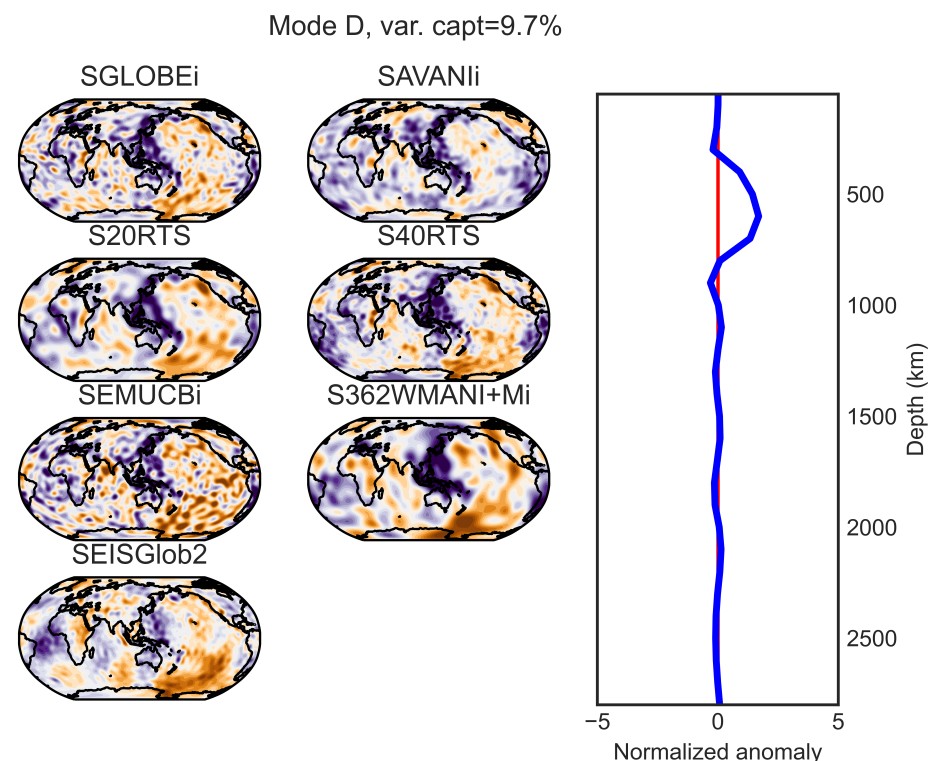

**Figure C4.** PC D (maximum at 600 km depth) of the combined analysis of the isotropic parts of the models.





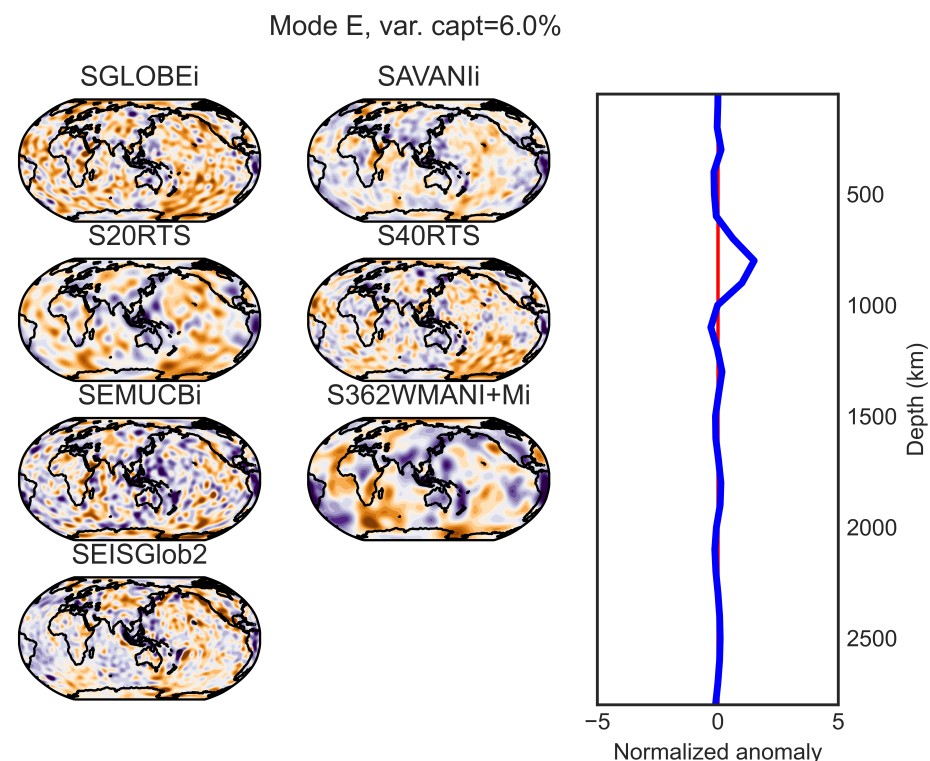

**Figure C5.** PC E (maximum at 800 km depth) of the combined analysis of the isotropic parts of the models.


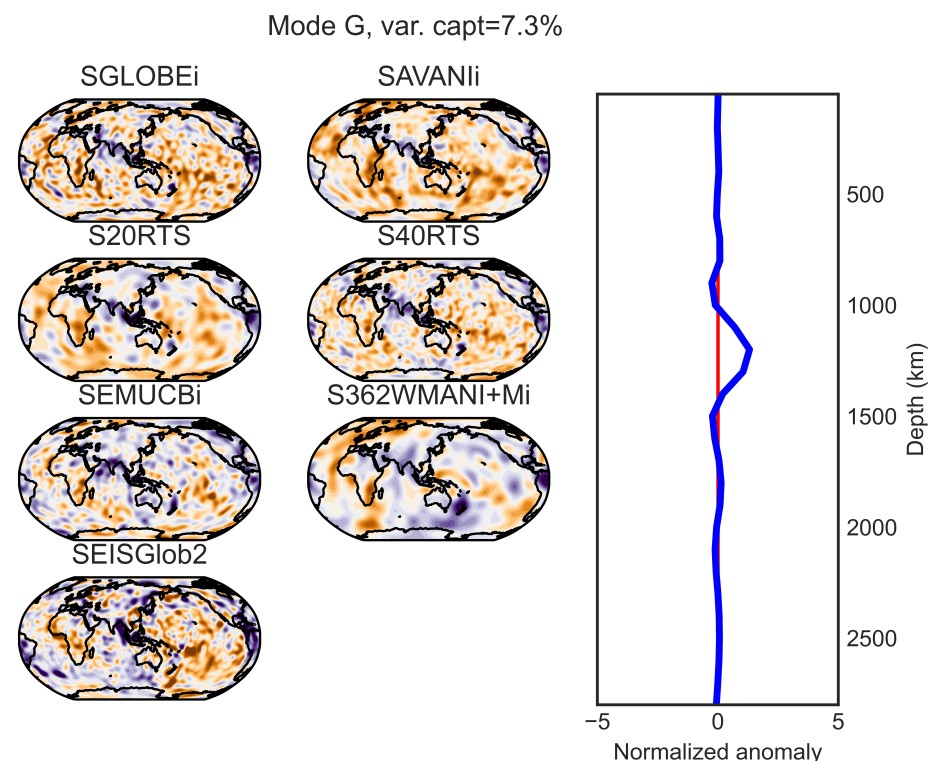

**Figure C6.** PC G (maximum at 1200 km depth) of the combined analysis of the isotropic parts of the models.



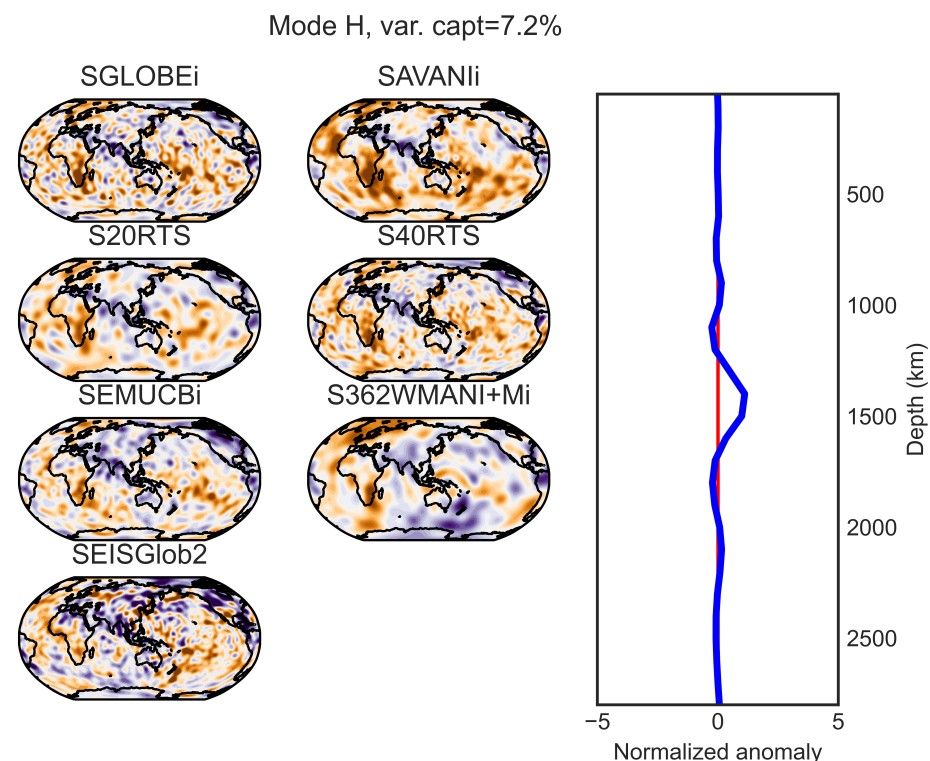

**Figure C7.** PC H (maximum at 1400 km depth) of the combined analysis of the isotropic parts of the models.





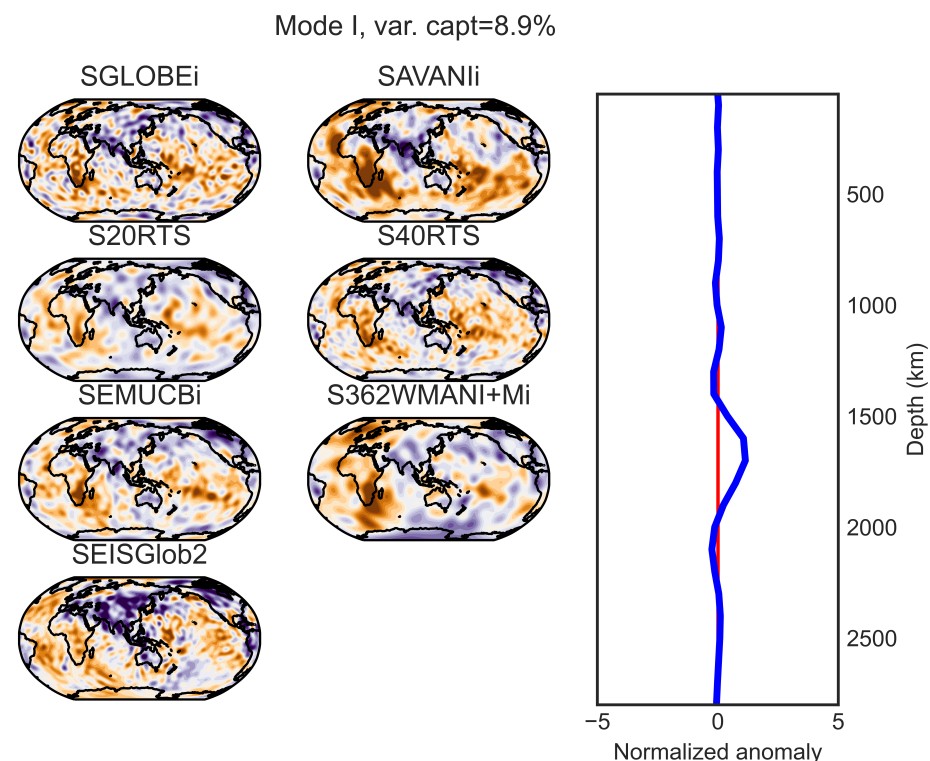

**Figure C8.** PC I (maximum at 1700 km depth) of the combined analysis of the isotropic parts of the models.


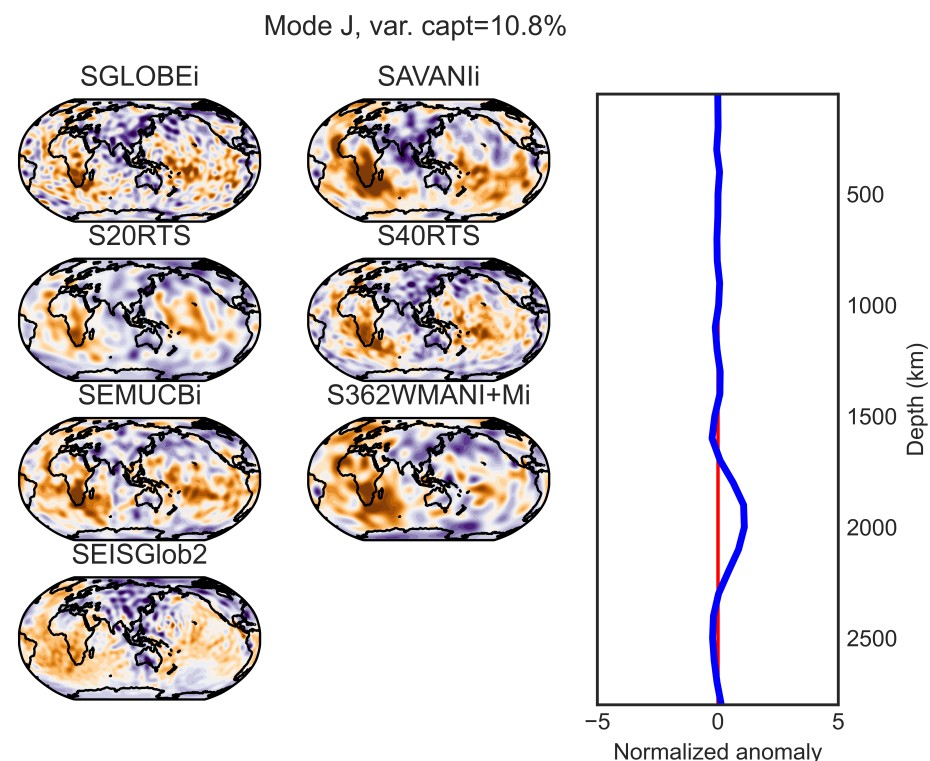

**Figure C9.** PC J (maximum at 2000 km depth) of the combined analysis of the isotropic parts of the models.





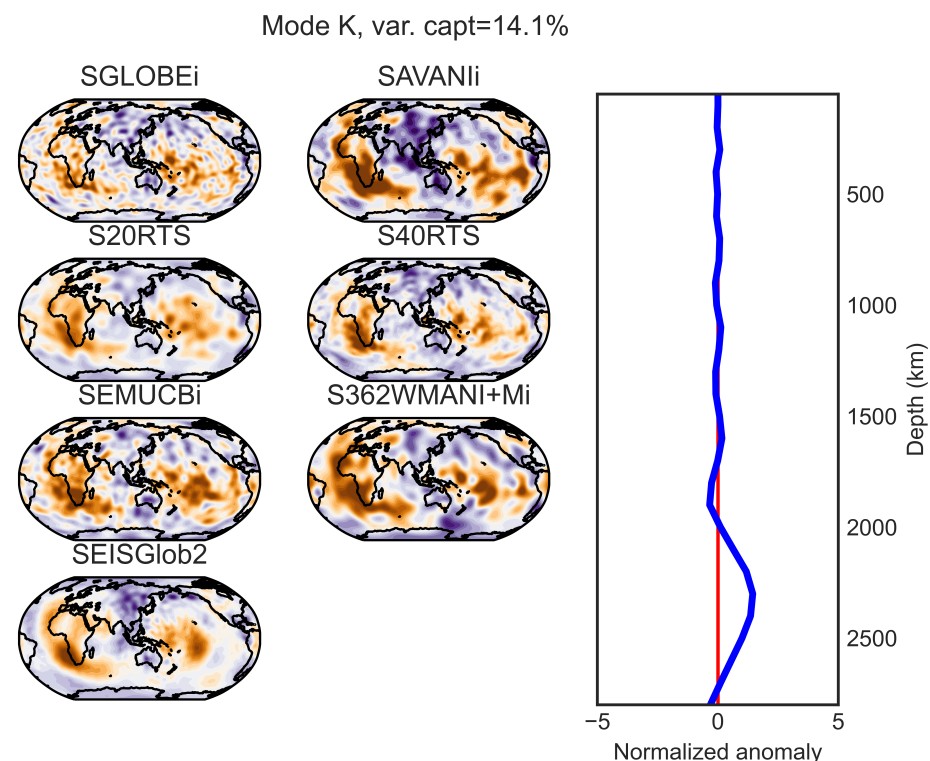

**Figure C10.** PC K (maximum at 2300 km depth) of the combined analysis of the isotropic parts of the models.





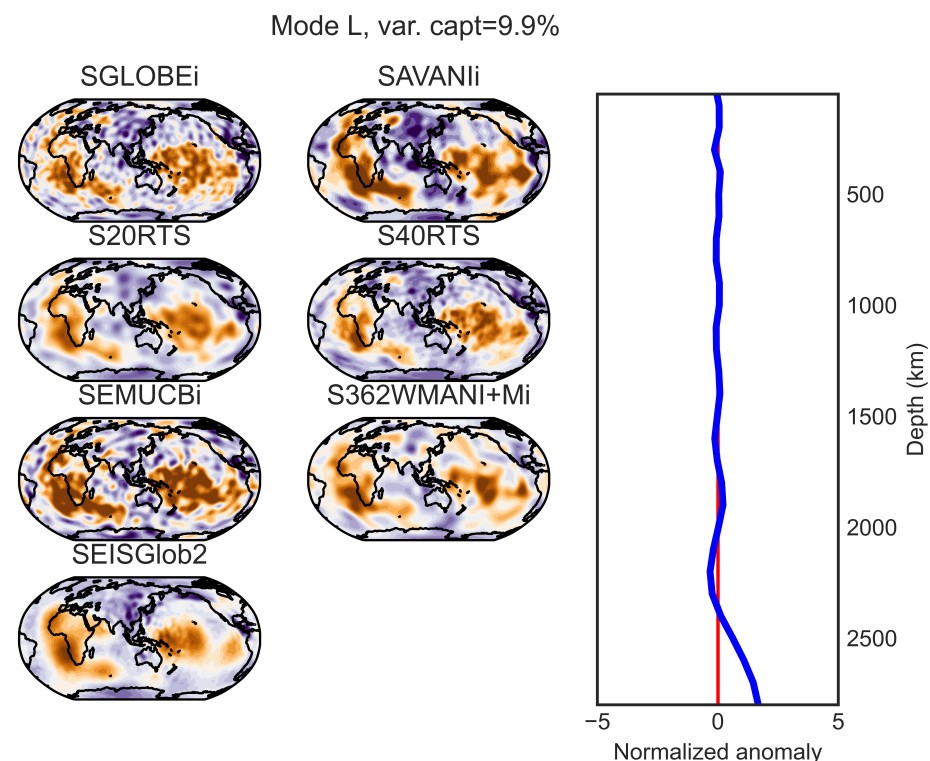

**Figure C11.** PC L (maximum at 2800 km depth) of the combined analysis of the isotropic parts of the models.



**Appendix D: Combined PCA of anisotropic models**

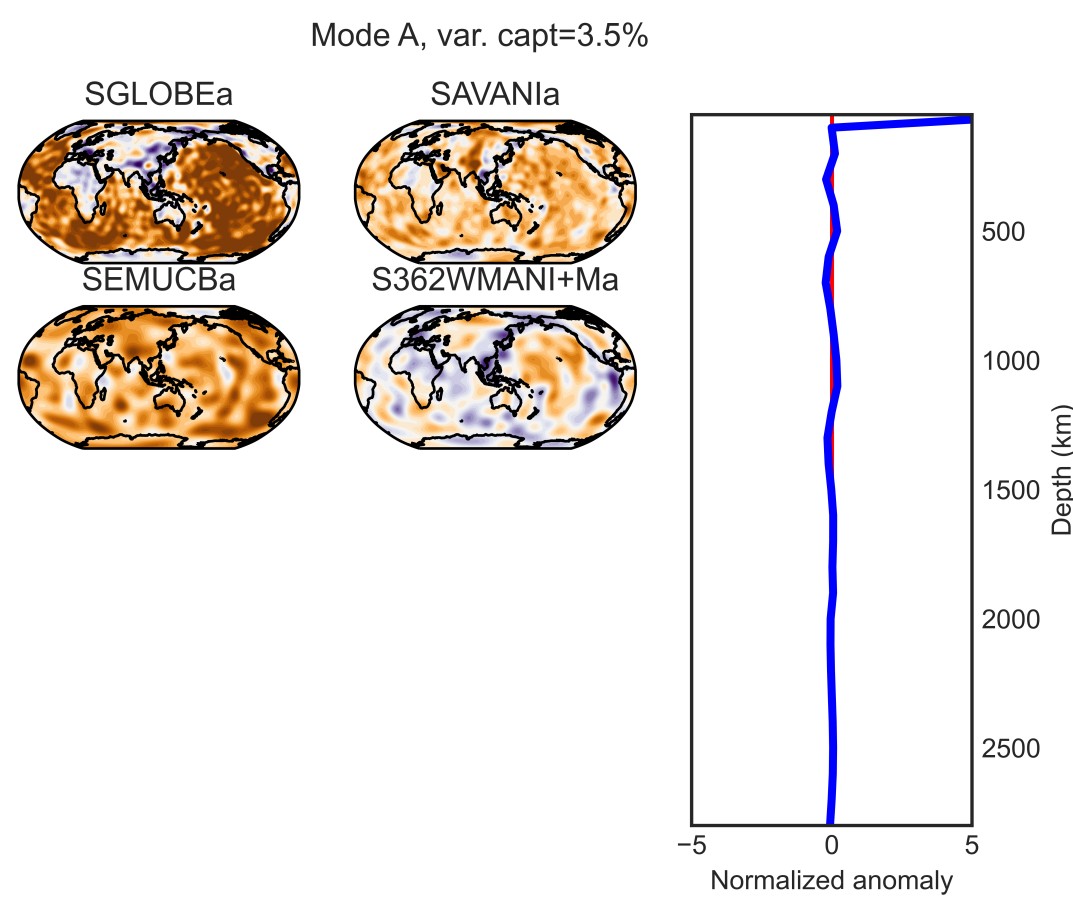

**Figure D1.** PC A (maximum at 50 km depth) of the combined analysis of the anisotropic parts of the models.




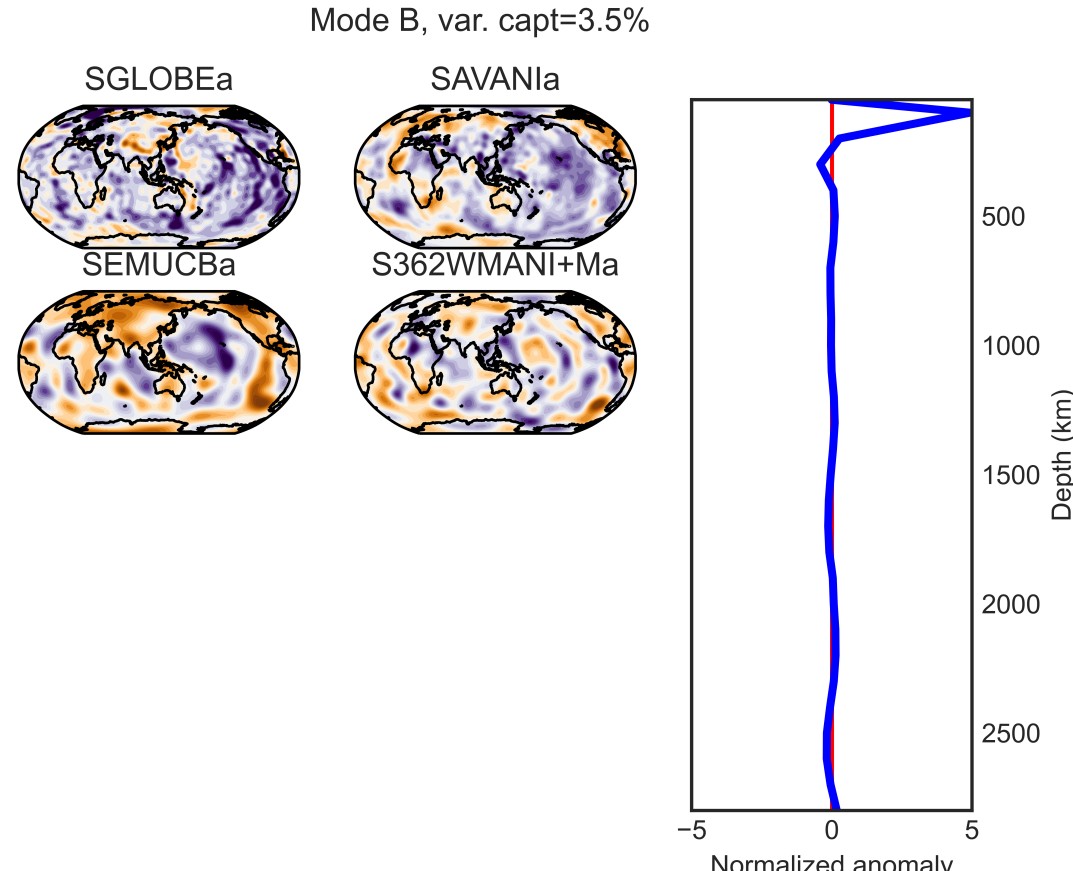

**Figure D2.** PC B (maximum at 100 km depth) of the combined analysis of the anisotropic parts of the models.




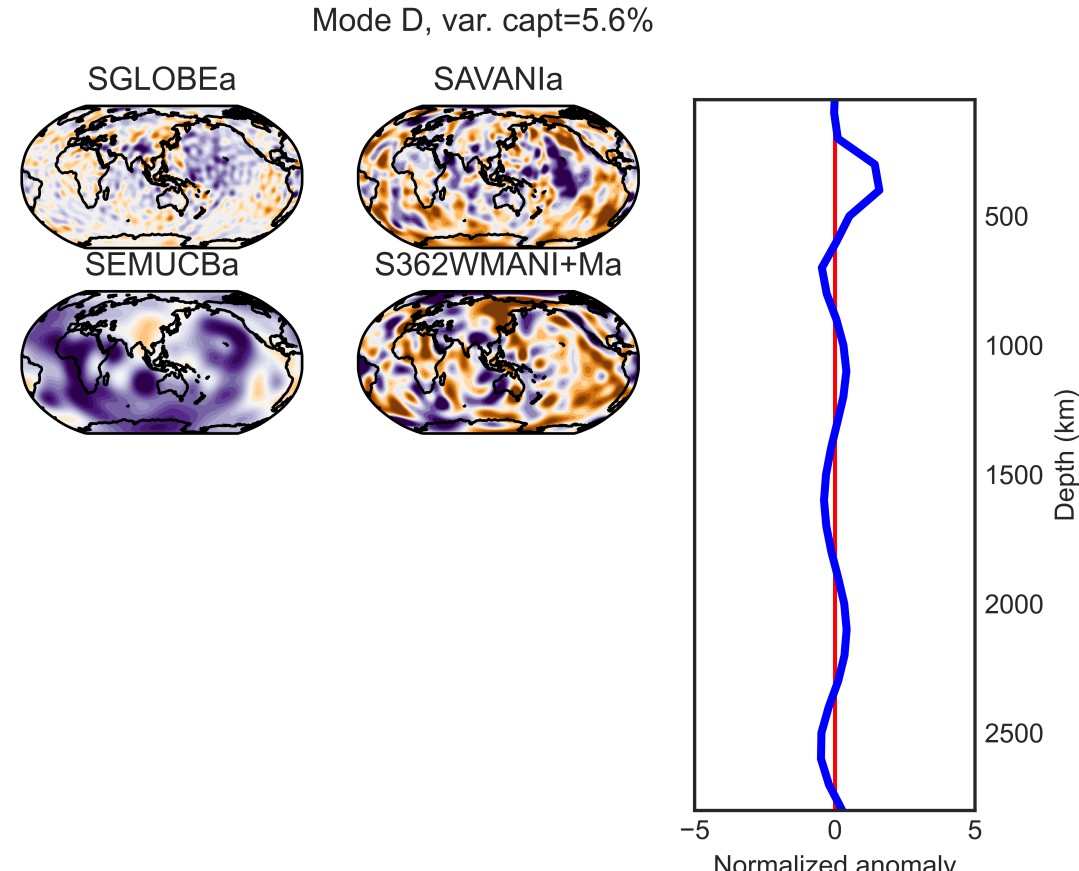

**Figure D3.** PC D (maximum at 400 km depth) of the combined analysis of the anisotropic parts of the models.





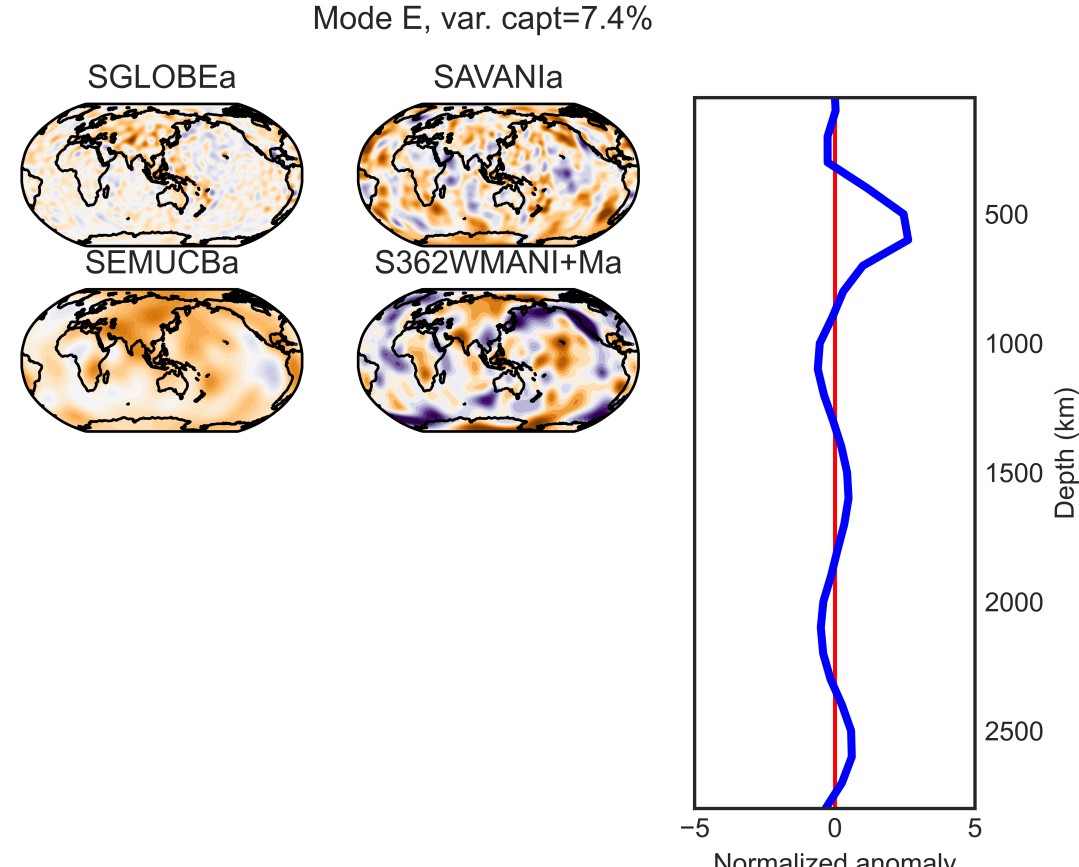

**Figure D4.** PC E (maximum at 600 km depth) of the combined analysis of the anisotropic parts of the models.





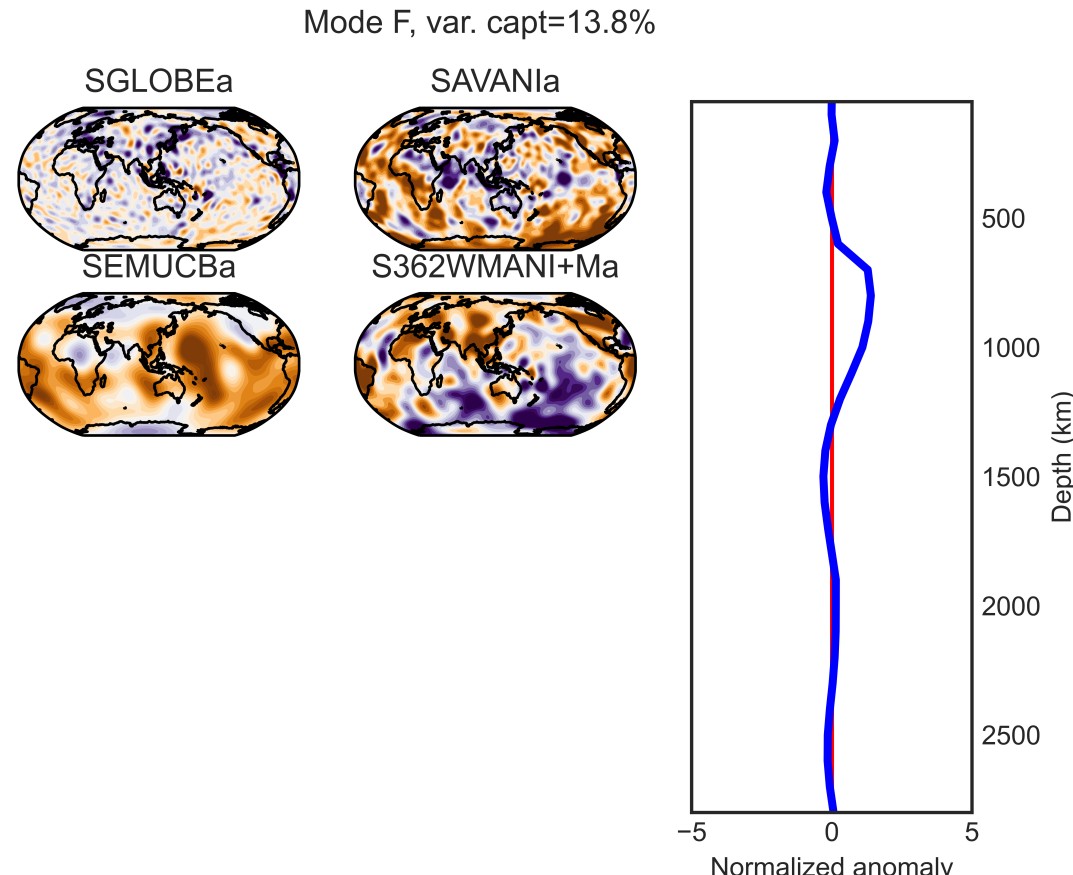

**Figure D5.** PC F (maximum at 800 km depth) of the combined analysis of the anisotropic parts of the models.





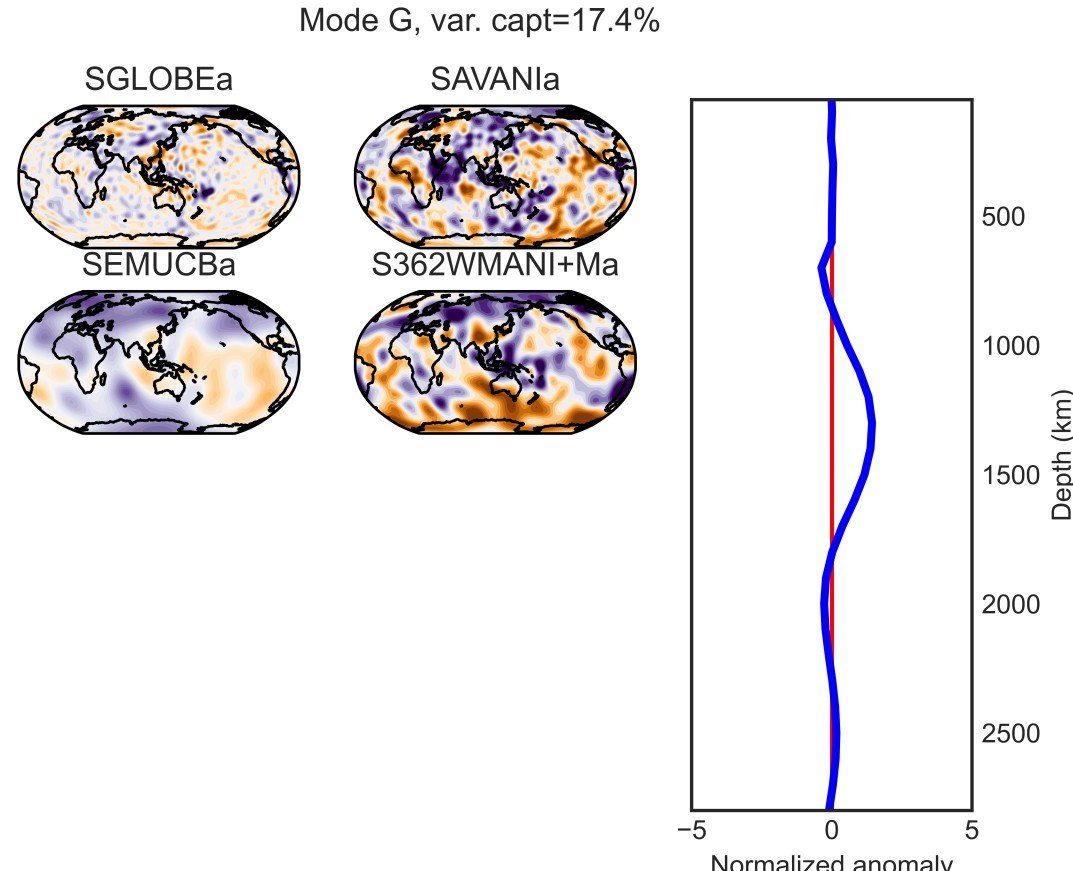

**Figure D6.** PC G (maximum at 1300 km depth) of the combined analysis of the anisotropic parts of the models.



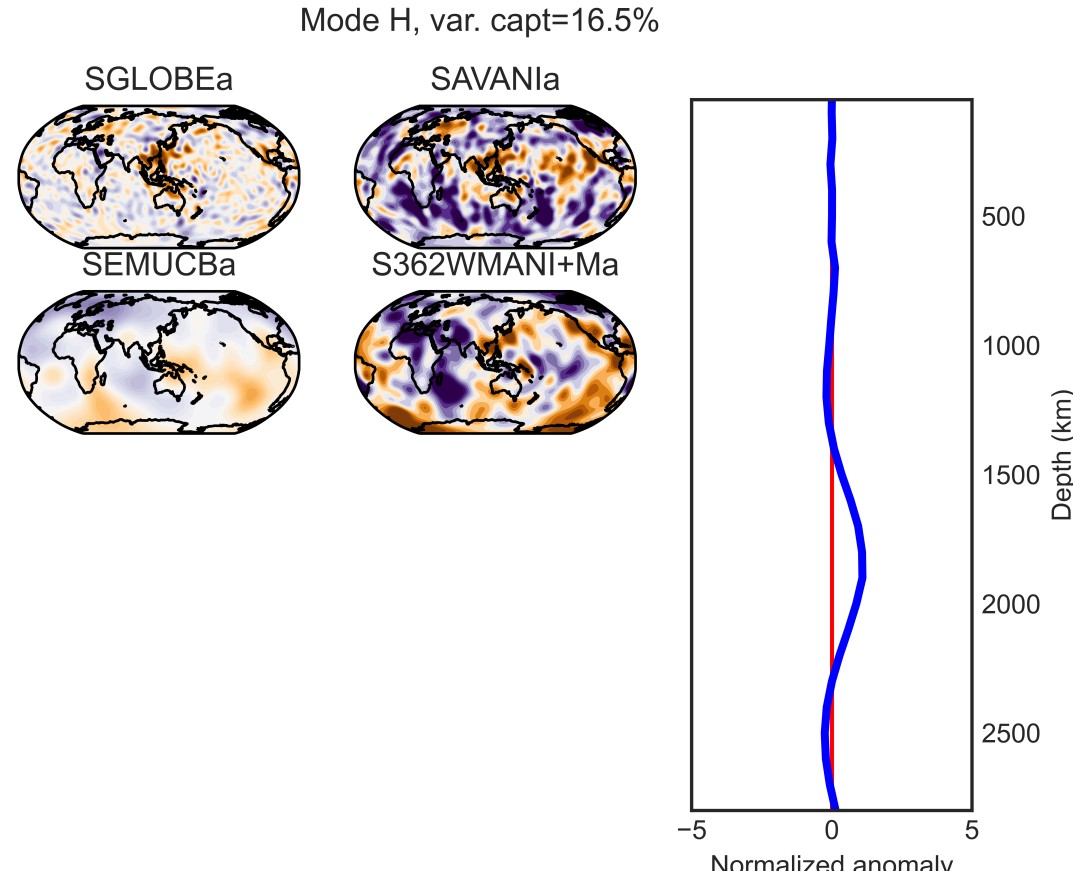

**Figure D7.** PC H (maximum at 1900 km depth) of the combined analysis of the anisotropic parts of the models.





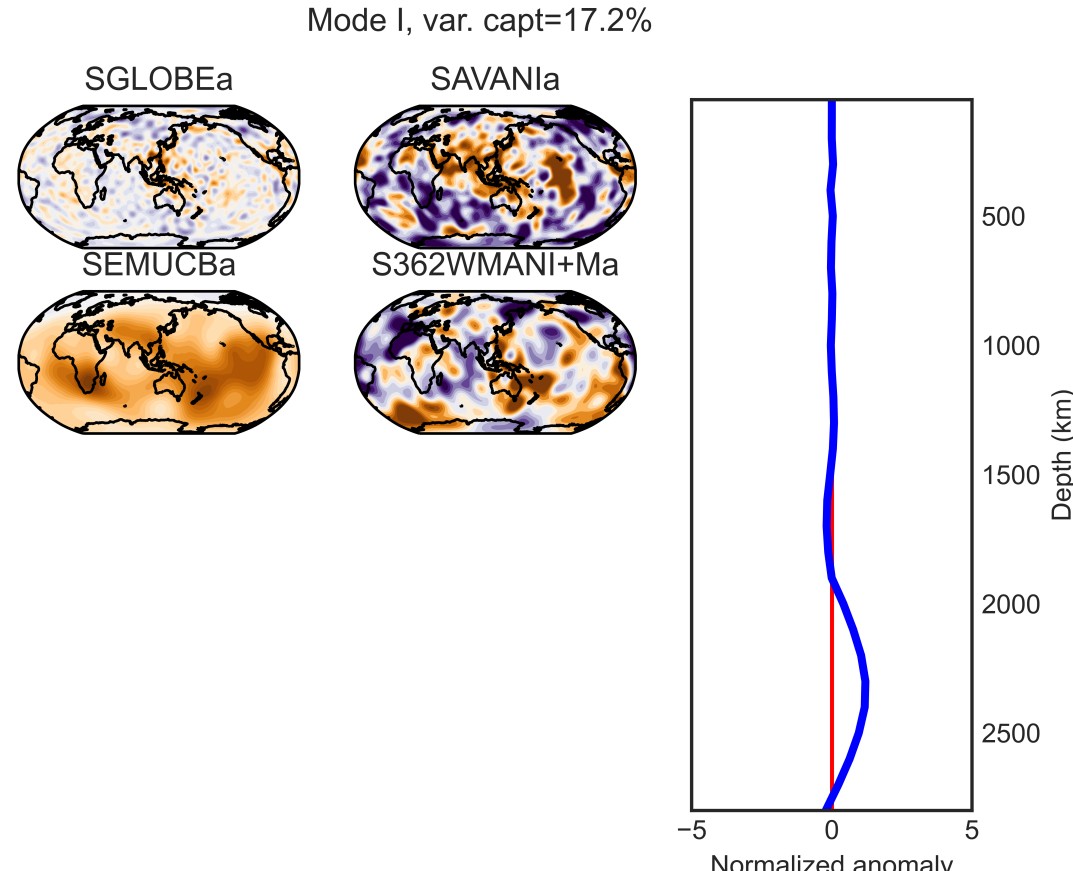

**Figure D8.** PC I (maximum at 2300 km depth) of the combined analysis of the anisotropic parts of the models.





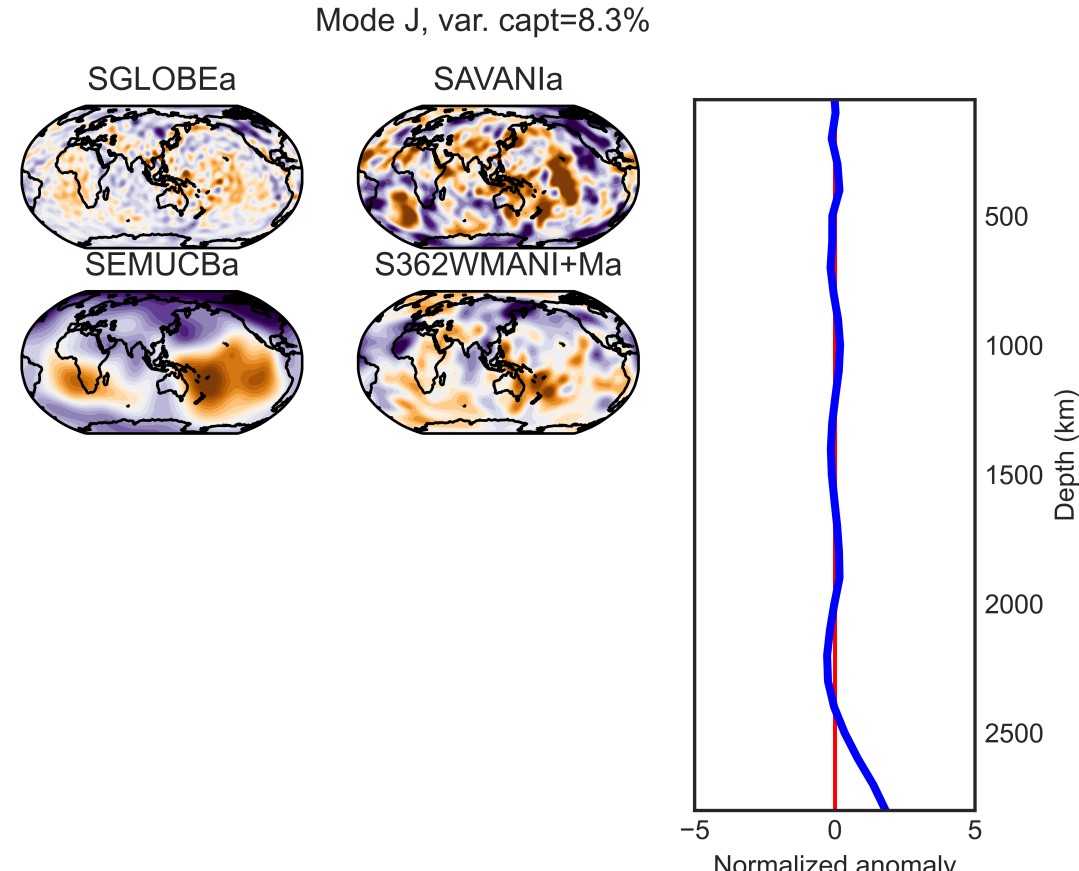

**Figure D9.** PC J (maximum at 2800 km depth) of the combined analysis of the anisotropic parts of the models.



*Author contributions.* OdV and AG performed the formal analysis, OdV and MVC developped and adapted the methodologies used, OdV made the visualization, all the authors contribute to investigation and validation of the results

*Competing interests.* The authors have no competing interests

*Acknowledgements.* A.M.G.F. thanks support from NERC Grant NE/N011791/1. We greatly thank Lapo Boschi, Barbara Romanowicz and Raj Moulik for providing us files with their models' parameterisations. OdV and AG work was financially supported by CNES as methodological development/test for analyzing and interpreting the geodesy mission data, in particular for GRACE satellite gravity. The two reviewers of a previous version of this manuscript are gratefully acknowledged for their insightful comments.





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
