# Peer review of "Comparing global seismic tomography models using the varimax Principal Component Analysis"

_Solid Earth, 2021_

## Author Response (AR1)

We are grateful to the reviewers for the time that they spent in their reviews and for their useful and constructive comments, which will greatly help us improve our manuscript.

We provide hereafter our answers in blue.

**Reply to review 1**

Generally, I find the method interesting; however, I am sceptical about its overall usefulness. Without a clear understanding/quantification of resolutions in these tomography models, I don't see why we need data compression techniques to decompose and then reconstruct the original tomography models (by considering some principal components). This may remove real signals from some tomography models (as only a limited number of components are considered). I understand that these methods, in general, can help to reduce the noise, but this requires a very good understanding of the resolution, as stated before. Nevertheless, this method can be used as a complementary method to other existing methods (such as k-mean clustering). On some existing platforms (e.g., SubMachine or IRIS EMC), different models are comparable by being projected on the same grid and corrected for different background models. Probably taking all these methods together will help us to better understand and compare these models.

We thank the reviewer for their comment, which made us realise that we did not explain well the whole purpose of the method. We do not consider varimax as a useful representation of the tomography models to be used instead of, e.g., the spline functions that were employed in the construction of some of the models. This would be useless. We rather see it as a diagnostic tool, allowing us to quantify the level of independent information in the tomography models and to compare the models more easily. Hence, we do not need a clear quantification of the resolution of the tomography models to perform the analysis.

We agree with the reviewer that the method is not the only one that can be used for this purpose, being complementary to existing methods and with the following advantages:

(1) Being data-based, it is neutral concerning the assumptions made in the model's construction (and, thus, we re-emphasise that we do not need a full quantification of the resolution to perform the analysis). For example, our simultaneous multi-model varimax analysis allows the comparison of the various tomography models on a neutral set of modes, determined by the level of compression fixed by the user.

(2) Based on the vertical consistency between the various tomography models, it provides a set of data-based vertical distribution functions. Those functions represent the information present in the model's reconstruction, and how the models relate to each other.

(3) It is fast and simple to implement, and, as we maximize the captured variance, the level of compression is lower for a given number of components/depths than would be required by other methods such as k-mean, for example (Figure 2).

(4) As the truncation level is a free parameter of the method, the user controls the amount of signal suppressed from the compression to an arbitrary level.

(5) As shown in the paper, it allows recovering all the Earth features discussed in the literature, allowing us to compare how each model captures and represents those features.

We modified our introduction and conclusion to make all these points clearer. See new text on L50-55; L428-434, L445-452.

**Specific comments**

1. I am worried that the couple of [%] which are not captured by the PCA/Varimax method contain crucial information and are not simply noise (e.g. L. 195). In my option, the small scale (and intermediate) structures/features are the interesting parts of the models to investigate. Hence, the method is suitable for looking into the large and intermediate scales but then why not "simply compare the models" to account for the small scale features? I'd appreciate that if the authors can explain this in the manuscript. This will significantly help the readers to understand the usefulness of the proposed method.

   The structure scale is not relevant for the varimax PCA method. It is only based on the vertical covariance. Considering the low amount of variance lost in the reconstruction (Figure 4) and the spectrum shown in the Supplementary Online Material, we capture most of the information, and we do not change the spectrum of the signal. So the method is valid for any scale, as long as the signal is robust. In the varimax comparison, what is called noise is not the small-scale features, but rather the part of the models that is not covariant vertically.

   We clarify this in the revised manuscript and hence that the analysis should not be removing any crucial information (see lines 428-434 of the revised manuscript).

   Of course, comparing the raw models would also be an option, but one has to determine which reconstruction (number of slices) is relevant for comparison. It is easier to compare 15 varimax components than e.g. more than 20 slices. In addition, the varimax method projects the models on a set of independent vertical profiles – which implies an optimal representation for a given number of slices (number of components) or a given amount of information (captured variance) - determined from the models themselves. This makes the varimax comparison optimal in the sense that you compare the largest amount of information for a given number of maps that are being compared.

2. I am not sure I understand the reasoning behind simplifying the tomographic model in the last step/after the inversion. The model is already a smooth/simplified version of the actual Earth structures. I think regularization is a safer method to remove "noise" from tomography models as it is, at least, informed by the measurements. The PCA/Varimax method does this as a "postprocessing". I am wondering why we need this postprocessing step?

   As we tried to explain above, the purpose of our study is not to represent or simplify tomography models but rather to provide an alternative, independent method to quantify the level of independent information in the tomography models, and to quantitatively compare different tomography models. With the varimax representation, we quantify the level of vertical independence of the information and how much of it is associated with the parametrization of the model – by comparing the varimax profiles with the splines or boxes of the model, for example. We find that this is a very useful tool to quantitatively compare how a given Earth feature is captured by the different tomography models.

   We modified the manuscript to clarify this (see L428-434 of the revised manuscript).

3. Maybe the easiest way to show the usefulness of this method is via synthetic tests? e.g., a synthetic global model that consists of large/intermediate-scale features as well as small-scale features. These small-scale features are generated via a stochastic process (noise), or they are real (but again, small-scale) structures. What would different components of PCA show? Can we reconstruct the small-scaled features?

A synthetic test may seem appealing at the first sight but we think that it is not useful for this particular study. If we generate a synthetic 3D global model with vertical consistency and add random noise on it, then the varimax method, being PCA based, will retrieve the model and diminish the noise, because this is what PCA is good at: separating the covariant part from the non-covariant one. If the noise is not covariant (vertically) it will disappear in the data compression. So, simply adding independent noise is not useful to teach us something that we do not already know about the quality of the varimax analysis. Adding real small-scale structures would also not be very useful because, as explained above, the varimax analysis is scale-independent, it depends only on the vertical covariance of the signal.

4. (This comment does not require any action/changes for this manuscript.) You describe in L.59 that you did not consider using P-wave models because the agreement is more limited between those models. New global P-wave models have a good agreement with existing S-models, also in the lowermost mantle. Maybe for your next analysis, you could consider including more "agreeable" P-models?

We thank the reviewer for this remark. We modified the manuscript (see lines 66-67 ) to explain that future work will expand the analysis to P-wave models, in light of recent work showing a better agreement between P-wave models.

In Figure 2, why does the PCA method show the LLVPs and the Varimax shows ridges and cratons. Why does it not capture structures at the same depth? What is the x-axis?

The PCA captures the most covariant vertical structures in the model, which correspond to the LLSVPs, given that they extend from the lower mantle nearly to 1000 km depth. The varimax keeps the same information as that retrieved by PCA, but it redistributes it between the components with maximal vertical compactness – the large values of the profiles are limited to as little depth as possible. That is why the two types of analyses capture structures at different depths. Both are valid representation/compression of the dataset, but the PCA presents artefacts from the domain geometry (In Richman review: the topographies of the PCs are primarily determined by the shape of the domain and not by the covariation among the data. In other words, different correlation functions on a geometrically shaped domain have similar Empirical Orthogonal Function (EOF) patterns in a predictable sequence, which do not reflect the underlying covariation. This is the case of square domains found on meteorological maps, or in our case, the progression of the unrotated PCs patterns is caused by a relationship between EOFs and harmonics). The varimax rotation captures distinct, well-defined depth domains in the mantle, which are easier to interpret physically.

The x-axis, i.e. the amplitude of the vertical eigenvectors, represents the maximum absolute value of the normalized anomaly at a given depth. It must be multiplied by the horizontal loading pattern, which provides normalized loads ranging between -1 and 1.

The manuscript is modified accordingly; also taking the suggestions of Reviewer 2 into account (see lines 137-144; 149-152 of the revised manuscript).

5.  Maybe I missed this part, but how do you handle different background models of the tomography models (e.g., PREM, IASP91, AK135, or even 3D background models)? How does this change PCAs? Is this being taken care of as you normalize the data (remove the mean and divide by standard deviation)?

As explained in lines 82-88 of the manuscript, we converted all the models into perturbations in shear wave speed and radial anisotropy with respect to the 1-D model PREM for fairer comparisons. Moreover, since both the vertical structure and the horizontal patterns are normalized, indeed, the background model does not impact the results.

We modified the manuscript to further clarify this point (see lines 86-88).
* * *
**Reply to review 2**

This manuscript applies principal component analysis (PCA) to identify and analyze patterns of structure found in global, seismic tomography models. The work presents a potentially useful tool to analyze commonalities and differences among the plethora of tomographic models that have been published over the years.

We thank the reviewer for the comments and suggestion for an improved manuscript.

1.  The use of PCA to identify main patterns of velocity variations in tomographic models is not entirely new (see Ritsema and Lekic, 2020), and I think the reader would benefit from an explicit comparison of the differences and similarities between this and past work.

We thank the reviewer for pointing us to the Ritsema and Lekic, 2020 study, we were not aware of it. To the best of our knowledge, that is the only previous study that used PCA to interpret tomography models, and we have now modified our manuscript to refer to it. Moreover, we also added some text discussing the differences and similarities between our analysis and that of Ritsema and Lekic, 2020 (see lines 34; 179-183 of the revised manuscript).

Such a comparison between k-mean, PCA and varimax was already present in the paper, in Section 4 and Figure 2. We discuss this comparison further in the new version of the paper and we now refer to Ritsema & Lekic (2020); see also our answer to comment 4.

2.  Nevertheless, this manuscript is complementary to and moves beyond this previous work in that it redistributes the principal components in a manner that concentrates them in-depth, using the varimax rotation. This is a clever and creative choice, which allows the method to identify patterns that can more directly be related to specific structures/target regions. Another interesting contribution is that the authors explore variations across tomographic models using a common set of PCs.

While a comparison of global tomographic models is a great place to start, I think the true power of this method might end up being in the analysis of local and regional tomographic models, which tend to use more diverse underlying datasets, and have highly variable spatial resolutions that could be revealed effectively by the type of PCA proposed here.

> We thank the reviewer for their constructive and positive comments. We agree that while comparing global tomography models is the best point to start, ultimately our method may indeed be useful for the interpretation and comparison of local and regional models. While such applications are beyond the scope of this study, which is already quite extensive, given that we are freely providing the codes used for the analysis, future analyses using local and regional models (as well as, e.g., geodynamical models) will be straightforward. We modified to text to explain this (see lines 465-468 of the modified manuscript).

3.    The authors point out that the number of PCs needed to explain nearly all (97.3%) of the variance in the tomographic models is always smaller than the number of splines/layers in the parameterization. Is this really surprising? After all, there is inherent smoothing of the retrieved structures due to both explicit regularization and data sensitivity. You write that "the splines or boxes seem to over-sample the available information" as if that is a bad thing. However, overparameterizing tomographic models and stabilizing the inversion through regularization has been advocated by some (e.g. Trampert and Snieder, 1996). It would be interesting to see which models and at what depths show the largest differences between the underlying parameterization width and that of the PCs.

> We agree that the fact that the number of PCs required is smaller than the number of splines is not a surprise. As they are based on splines or boxes, there are always fewer PCs than splines or boxes by construction. We did not mean the over-sample as a bad thing; rather, we want to point out that the PCA objectifies the number of layers/splines required considering the information at hand.

> We compare the splines and varimax-PCs in Figure 5 of the submitted manuscript (Figure 3 of the new version). We can see that except for the boxes from SAVANI, which have no relations with the PCs, all the models have a few splines that roughly coincide with PCs, and a majority of PCs seemingly independent from the splines. The PCs close to splines are located in the upper part of the lower mantle. This is discussed in line 230-235 (or 247-251 in the new version). But, we agree that the use of the expression "over-sample" in the conclusion is confusing.

> Instead of "...where the splines or boxes over-sample the available information", we wrote: "where the information is recovered by fewer PCs than the number of spline functions or boxes" (see lines 265 of the revised manuscript).

4.    Doesn't the normalization applied to the velocity variations (standardization at each depth) skew the analysis toward the large mid-mantle areas in which velocity variations are quite small and often poorly resolved? Relatedly, the power in each component seems to be much smaller in this study than in Ritsema and Lekic (2020). Could this be because they did not normalize by depth?

The normalization does affect the result, giving more weight to the areas with smaller variations than what one would get without normalization. On the other hand, without normalization, most of the modes focus on the upper mantle, where the variations are larger. This is probably the case for the Fig. 4 of Ritsema and Lekic (2020) (R&L):

1)      For both models tested in R&L, their first PC is driven by the shallowest zone (tectonic pattern), while in our analysis (e.g., S40RTS) the 1st PC is driven by the LLSVPs (Fig. 2 in our original ms);

2)      Due to the absence of normalization, even the three other PCs of R&L contain a lot of energy in the shallowest zone, while our normalization allows keeping the tectonic-driven zone in essentially two components out of 6 (#4 capturing 7.6% of the variance and #5 capturing 6.2% on our Fig 2, in the top panels corresponding to PCA). When working without normalization, as in Ritsema & Lkic (2020), all our 6 eigenvectors show a strong contribution from the shallowest mantle (< 250 km), where most of the variance is located.

Considering the uncertainties about the middle mantle, we consider it interesting to have a fair amount of modes in that region, allowing a better model comparison and more readability.

These points and the effect of using a normalization are clarified in the new version of the manuscript (see lines 179-186).

5.     The authors identify differences in the number of PCs that are required to compress tomographic models, and attribute them to differences in regularization. That is certainly a key parameter, but another difference among the models that is worth discussing pertains to what kind of data are included in the analysis. For example, if the model does not include constraints from overtones, structures in the transition zone and mid-mantle might not be well-retrieved. Treatment of discontinuity topography can also matter because neglecting this topography can map directly into isotropic wave speed variations in the mid-mantle.

We fully agree with this suggestion and have now modified the discussion in the manuscript accounting for the kinds of data used to build the different models (see the text in lines 270-273 of the modified manuscript).

6.     I would prefer to see a summary figure of some kind, that synthesizes the information that is currently presented across many panels. I was keen to look at all the panels, but most readers might not be, and they would certainly appreciate a figure that eliminates the need for them to make comparisons between panels on their own. Overall, the number of figures/panels in this manuscript can be overwhelming.

We do not think that the existing figures can be further compressed without a (considerable) loss of readability. Nevertheless, considering that the number of figures might jeopardize the manuscript's legibility, we moved some figures to the supplementary material:

·Figure 1, which summarizes published information;

·Figure 4, as the explanation in the main text seems sufficient to make our point.

·Figure 8, as the previous figures on the isotropic models, are sufficient to illustrate the relevance of the varimax method, considering that the result with the anisotropic method is not very convincing.

This increases the number of figures in the supplementary information, but we prefer to keep all that information for completeness and reproducibility of our work. This limits the figures in the main paper to five, making it more concise and hopefully more pleasant to read.

7.    I also had some minor questions and comments that the authors might wish to respond to and address:

● If the goal is compression, why use the varimax rotation? It is my understanding that the original PCs would always provide better compression.

   The compression is exactly the same for varimax and PCs, whereas the PCs have several drawbacks, as explained in the paper.

   We added a sentence to clarify this point (see lines 138-144; 148-150). See also our answer to Reviewer 1 where we provide some more details on the main rationale for applying the varimax method.

● When discussing the patterns of heterogeneity, I kept thinking back to the term "heterosphere" introduced in Dziewonski et al. (2010) to describe the strong seismic heterogeneity in the tectonic uppermost 250 km of global tomographic models.

   We thank the reviewer for this useful point and we now refer to the "heterosphere" in the manuscript (see lines 287-288)

● The 1D reference model for S362WMANI+M should be the STW105 model of Kustowski et al. 2008.

   We thank the reviewer for spotting this mistake, which is now corrected.

● When comparing countable quantities (like the number of PCs), you should use the word "fewer" rather than "less", etc.

   Corrected.